



# A kernel-driven BRDF model to inform satellite-derived visible anvil cloud detection

Benjamin Scarino[1], Kristopher Bedka[2], Rajendra Bhatt[1], Konstantin Khlopenkov[1], David R. Doelling[2], and William L. Smith Jr.[2]

[1]Science Systems and Applications, Inc., One Enterprise Pkwy Ste 200, Hampton, VA 23666 USA
[2]NASA Langley Research Center, 21 Langley Blvd MS 420, Hampton, VA 23681-2199 USA

*Correspondence to*: Benjamin R. Scarino (benjamin.r.scarino@nasa.gov)

**Abstract.** Satellites routinely observe deep convective clouds across the world. The cirrus outflow from deep convection, commonly referred to as anvil cloud, has a ubiquitous appearance in visible and infrared (IR) wavelength imagery. Anvil clouds appear as broad areas of highly reflective and cold pixels relative to the darker and warmer clear sky background, often with embedded textured and colder pixels that indicate updrafts and gravity waves. These characteristics would suggest that creating automated anvil cloud detection products useful for weather forecasting and research should be straightforward, yet in practice such product development can be challenging. Some anvil detection methods have used reflectance or temperature thresholding, but anvil reflectance varies significantly throughout a day as a function of combined solar illumination and satellite viewing geometry, and anvil cloud top temperature varies as a function of convective equilibrium level and tropopause height. This paper highlights a technique for facilitating anvil cloud detection based on visible observations that relies on comparative analysis with expected cloud reflectance for a given set of angles, thereby addressing limitations of previous methods. A one-year database of anvil-identified pixels, as determined from IR observations, from several geostationary satellites was used to construct a bidirectional reflectance distribution function (BRDF) model to quantify typical anvil reflectance across almost all expected viewing, solar, and azimuth angle configurations, in addition to the reflectance uncertainty for each angular bin. Application of the BRDF model for cloud optical depth retrieval in deep convection is described as well.

## 1 Introduction

Satellite imagery offers a valuable perspective for tracking deep convection that is advantageous in being both spatially broad and contiguous, and also persistent in time. Deep convective clouds (DCC) are comprised of one or more updraft regions, some of which penetrate into the lower stratosphere and are referred to as overshooting tops (OTs), and cirrus outflow emanating from the updrafts that are referred to as anvil cloud. Deep convection appears as spatially coherent cold and highly reflective regions in infrared (IR) and visible imagery (Kirk-Davidoff et al. 1999; Shindell 2001; Setvák et al. 2010; Homeyer and Kumjian 2015; Bedka et al. 2015; Bedka et al. 2016). Automated satellite-observed DCC detection based on recognition of these patterns is important for a variety of reasons. Deep convective clouds generate hazardous weather conditions such as heavy rainfall, lightning, aviation turbulence and icing, damaging wind, hail, and tornados (Bedka et al. 2010; Bedka and Khlopenkov 2016; Yost et al. 2018). Forecasters can benefit from satellite-based guidance that can identify these hazardous weather conditions (Gravelle et al. 2016a; Gravelle et al. 2016b). Deep convective clouds are also a common Earth target used for vicarious satellite instrument calibration (Doelling et al. 2013; Doelling et al. 2016; Bhatt et al. 2017a; Bhatt et al. 2017b; Doelling et al. 2018). Furthermore, researchers studying upper troposphere and lower stratosphere composition benefit from knowing where DCC and OT occurred for use in trajectory models (Herman et al. 2016; Smith et al. 2017; Vernier et al. 2018).


The human eye is rather adept at identifying patterns indicative of deep convection – easily being able to locate coherent and circular or elliptical regions of bright, cold, and persistent clouds. Replicating a human-like recognition approach can be problematic, however, because solar illumination and viewing geometry variations affect the appearance of deep convection. That is, a basic pattern recognition algorithm may miss or falsely detect DCC depending on time of day, the angle at which the

clouds are viewed, and tropopause temperature. For example, Figs. 1a and 1b show the calibrated visible (VIS) reflectance of the same mesoscale convective system (MCS) over the Texas panhandle as viewed from GOES-West (GOES-15) and GOES-East (GOES-13), respectively, on 25 May 2015. Despite spectral band normalization and having the same relative calibration reference of Aqua-MODIS, the GOES-West view shows significantly higher reflectance values due to being in the forward-scatter position at 12:30 UTC (Scarino et al. 2017; Doelling et al. 2018). Later, at 23:45 UTC when GOES-East is in the forward-

scatter position, the GOES-East view of the mature MCS now appears to be the brighter of the two images (Fig. 1c and 1d). It is easy to see, therefore, how a DCC mask based on a simple reflectance threshold would mischaracterize this storm depending on whether it was viewed from the east or west. The appearance of a storm can vary significantly owing only to illumination and/or viewing conditions, and therefore a justifiable need exists to carefully quantify the expected anisotropic reflectance before visual-based judgements of DCC can be reliably made.

The increased interest in the development of automated, geostationary- (GEO) satellite-sourced means of severe weather and climate analysis in recent years necessitates continual advancement in skillful identification of DCC. Bedka et al. (2010), relied on fixed temperature thresholds based on longwave IR window (~11 μm) observations, which often resulted in seasonal and regional biases. Bedka and Khlopenkov (2016) employed better IR pattern recognition and the addition of a VIS-channel-based (~0.65 μm) algorithm that emulates processes used by humans to identify DCC top anvils and their embedded OTs, which

yielded significantly improved detection consistency and quality. The product is valuable for purposes of operational forecasting of severe weather, especially in regions without adequate contiguous weather radar coverage, whether due to isolation, terrain, or influence of national borders. Continued quantification of algorithm accuracy and validation relative to other remotely sensed datasets are key to product development.

Other methods have been introduced that attempt to objectively recognize DCC features. A common technique is the well-

documented multispectral band water vapor (WV) minus IR (WV−IR) brightness temperature (BT) difference (BTD) method (Schmetz et al. 1997; Setvák et al. 2007; Martin et al. 2008). Ai et al. (2017) demonstrated that AIRS WV−IR BTD can effectively identify overshooting tops if the WV-channel wavelength is located in a very strong WV absorption band. Although this approach can be effective, it relies on the presence of a WV channel, which can have significant spectral variation across the global constellation of geostationary imagers or may be absent entirely on some imagers. For broad applicability, many methods

often rely on fixed single-channel BT thresholds, often in the range of 195 K to 225 K, with the threshold dependent on the product application, method, and satellite (Bedka et al. 2010; Young et al. 2012; Doelling et al. 2013; Bhatt et al. 2017a; Bhatt et al. 2017b; Doelling et al. 2018). One drawback of relying on such thresholds is the zonal dependence of cloud-top IR BT. That is, severe storms at mid-to-high latitudes will have warmer cloud tops than comparably severe storms at low/tropical latitudes. Figures 2a and 2b illustrate this point, showing the IR BT of two groups of severe-weather-producing storm systems on 31

August 2018 at 20:47 UTC – one group at higher latitudes of the Contiguous United States (CONUS) over Minnesota, and the other group at lower CONUS latitudes over the Gulf of Mexico and southeastern states. The coldest IR BT value found in Fig. 2a is near 207 K, whereas Fig. 2b shows IR BT approaching 196 K in many areas. As such, a 205-K IR BT threshold (see DCC-based vicarious calibration methodologies of Doelling et al. and Bhatt et al.), for example, would classify abundant DCC in the southern latitudes, but no DCC at the higher latitudes, despite reports of 2-3-inch (~5.0-7.5-cm) hail in Minnesota. If one

normalizes for tropopause height, by computing the IR minus tropopause (IR−Trop) BTD, the apparent intensity of the storms in





the north (Fig. 2c), especially the pronounced cell near the western central Minnesota border that produced the severe hail, is more comparable to that of the southern storms (Fig. 2d). Note that the aforementioned vicarious calibration approaches that employ a fixed 205-K threshold are limited to tropical latitudes, and thus suffer no consequence for lack of tropopause normalization (Doelling et al. 2013; Bhatt et al. 2017a; Bhatt et al. 2017b, Doelling et al. 2018). For broader study areas,

however, the Fig. 2 example demonstrates the importance of consideration, and potentially normalizing, for one's analysis environment when developing a globally applicable approach to DCC characterization.

Although the OT/anvil detection technique developed by Bedka and Khlopenkov (2016) works reasonably well, the VIS component of the two-channel methodology suffered from a lack of information on expected anvil cloud reflectance for given viewing, solar, and azimuthal angle combinations. Visible-wavelength satellite observations have been routinely used to develop

models for bidirectional reflectance distribution function (BRDF) over non-Lambertian land surfaces, which is useful for angle-dependent pattern recognition through provision of expected reflectance values to which measurements are compared. With the proper atmospheric correction (Hu et al. 1999; Lucht et al. 2000; Radkevich 2018), the high-resolution and varied angular sampling retrieval capabilities of instruments like the Moderate Resolution Imaging Spectroradiometer (MODIS), the Multiangle Imaging Spectrometer (MISR), the polarization and directionality of Earth reflectances (POLDER) radiometer, and the

Advanced Very High Resolution Spectrometer (AVHRR), have yielded reliable operational land surface BRDF and albedo products for nearly two decades, with significant benefit to the research community (Lucht et al. 2000; Schaaf et al. 2002; Jin et al. 2003; Huang et al. 2013; Vasilkov et al. 2017). These high-resolution retrieval capabilities also extend to modern, advanced GEO satellite imagers, such as that from Himawari-8, thereby granting the benefit of high temporal frequency to multi-angular pattern-analysis-based BRDF modeling, i.e., a kernel-driven concept (Matsuoka et al. 2016). A similar kernel-driven BRDF

model technique based on DCC may therefore help normalize VIS-imagery-based anvil cloud identification across almost all illumination and viewing conditions.

Although DCC anvils are, in certain conditions, of the most Lambertian Earth reflectance targets, a BRDF correction is necessary to properly characterize cloud-top surface scattering as a function of illumination and viewing geometry, especially for larger angles (Bhatt et al., 2017b). Unlike the case of land surface BRDF retrieval, which requires atmospheric correction, DCC

tops reside near the tropopause, above which absorption effects are relatively small and thus albedo distribution is assumed to be effectively constant month to month (Hu et al. 2004). Owing to these characteristics, a DCC-based VIS BRDF model was developed from Clouds and the Earth's Radiant Energy System (CERES) and the Visible Infrared Scanner (VIRS) observations for the purpose of post-launch calibration of satellite sensors. The main premise of this vicarious calibration approach is that the distribution of DCC albedo remains stable in time, and any temporal shift observed in the DCC reflectance distribution can be

attributed to satellite sensor degradation (Hu et al., 2004, Doelling et al., 2013). Bhatt et al. expanded this DCC calibration technique to shortwave infrared channels by constructing channel-specific seasonal BRDFs from Suomi National Polar-orbiting Operational Environmental Satellite System (NPOESS) Preparatory Project (SNPP) Visible Infrared Imaging Radiometer Suite (VIIRS) observations and applying the result to the corresponding MODIS bands. The DCC technique allows for characterization of sensor gain stability early in an instrument's lifetime – forgoing the two-year time period necessary for

traditional deseasonalization methods, and thereby enabling more-timely calibration stability analyses for any imager with a similar sun-synchronous orbit (Bhatt et al. 2017a; Bhatt et al. 2017b). These studies demonstrate that a DCC-sourced BRDF can accurately predict expected cloud reflectance for a given range of viewing zenith angle (VZA), solar zenith angle (SZA), and relative azimuth angle (RAA) conditions, thereby allowing for accurate monitoring of satellite imager stability. Expanding such a technique for cloud reflectance prediction to GEO satellites can aid inter-consistency studies and benefit anvil cloud detection.



This article proposes a new kernel-driven BRDF model, which finds its application in enhancing anvil cloud detection (and thereby OT detection) capability and cloud optical depth (COD) parameterization. We will describe the satellite-derived data used to formulate the model, as well as explain the development and uncertainty of an anvil reflectance prediction look-up-table (LUT), which shapes the BRDF model. We will show that the kernel-driven approach provides reasonable estimates of expected

reflectance for widely varying solar illumination and viewing conditions, thereby promoting consistent identification of anvil cloud regardless of time of day or satellite view. Furthermore, a BRDF of expected reflectance for any viewing condition allows for a quick parameterization of COD based only on the difference between observed reflectance and the model prediction. The parameterization is developed based on multispectral retrieval employed within the NASA Langley Research Center SATellite ClOud and Radiation Property Retrieval System (SatCORPS) framework in support of the CERES project, in which GEO cloud

retrieval relies on the CERES Edition 4 algorithm for global cloud detection (Trepte et al. 2019; Minnis et al. 2020). The timeliness and consistency of the anvil detection scheme and related COD parameterization owed to the BRDF model are beneficial to operational forecasting and nowcasting efforts, e.g., convection avoidance or interception for flight routing purposes or airborne science campaigns, which rely on accurate, real-time information.

## 2    Data and Methods

### 2.1 Geostationary satellite imagery

A twelve-month database of GOES-13, GOES-15, and Himawari-8 VIS and IR satellite imagery from December 2016 through November 2017 was used to develop the BRDF model. This time period was chosen such that a full seasonal cycle is characterized without influence of orbital shifts, which did occur for GOES-13 when it was replaced by GOES-16 in December of 2017. Half-hourly observations are acquired from sunrise to sunset between 65° N and 65° S, and from 130° W to 30° W for

GOES-13, 175° E to 90° W for GOES-15, and 88° E to 178° W for Himawari-8, combining all partial hemisphere, hemisphere, and full disk scanning patterns. It is assumed that using observations across four seasons, high latitudes, diverse regions, and to solar terminator will yield a reasonably full range of possible VZA, SZA, and RAA combinations for observed anvil clouds, with significant statistical population – serving as a strong empirical foundation to shape the BRDF model. All geostationary imagery was acquired from the University of Wisconsin-Madison Space Science and Engineering Center (SSEC).

The 0.5-km VIS and 2-km IR Himawari-8 nadir spatial resolution scales were resampled to 1 km and 4 km, respectively, in order to better match those for GOES-13 and GOES-15. The VIS retrievals are subsampled to the IR data resolution, so therefore the final output resolution is 4 km for all parameters. The resampling process typically preserves the BT signal of anvil clouds, which ideally are relatively homogenous across many kilometers. Resampling also acts to dampen any potential small-scale VIS and BT variability in convective anvils that may otherwise influence construction of the model. The resampling function, which

is based on Lanczos filtering with the parameter $a=3$ extended to the two-dimensional case, is applied over an array of $6 \times 6$ pixels, which is padded with replicated edge-pixel values near image boundaries (Duchon 1979). This interpolation method is based on the sinc filter, which is known to be an optimal reconstruction filter for band-limited signals such as digital satellite imagery (Bedka and Khlopenkov 2016).

Relative consistency of reflectance observations between the three instruments is ensured by application of CERES Edition 4

VIS imager calibration coefficients for each GEO, which are determined from the monthly gain trends of GEO and Aqua MODIS spectrally-consistent, ray-matched radiance pairs over all-sky tropical ocean, DCC, and invariant desert scenes, based on the best-practices of the Global Space-based Inter-Calibration System (GSICS), and with uncertainty less than 1% (Goldberg et al. 2011; Scarino et al. 2017; Doelling et al. 2018). Furthermore, although relative consistency of BT values is not necessary to




develop the BRDF model, IR calibration is based on hourly adjustments to GSICS-referenced VIRS ray-matched pairs (Scarino et al. 2017).

The Meteosat Second Generation (MSG) satellites are not included in this analysis because 1-km VIS imagery is not collected across the entire 65 °N to 65 °S domain throughout a day. Much of the Northern Hemisphere is observed at 1 km but, over the

Southern Hemisphere, a moving window of 1-km data is collected, which follows the Sun and captures data during well-illuminated periods of the day. Visible data are only available at 3-km resolution across the full disk view, which would be inconsistent with the GOES and Himawari-based analyses. Given that MSG data is incorporated into the GSICS inter-calibration analysis, we expect that methods developed from GOES and Himawari will perform consistently when applied to MSG data. Note that for some analyses, satellite data are supplemented by modeled atmospheric profiles provided by the Global Modeling

and Assimilation Office (GMAO) Modern-Era Retrospective analysis for Research and Applications, Version 2 (MERRA-2) product.

## 2.2 Multi-angle lookup table for anvil cloud reflectance

A three-dimensional lookup table (LUT) of anvil cloud reflectance was built from pixels classified as anvil using IR observations that satisfy a set of conditions (Bedka and Khlopenkov 2016; Khlopenkov and Bedka 2018). The specifics of the IR-based anvil

classification process are described in Appendix A. The three dimensions of the LUT are VZA, SZA, and RAA, yielding the mean anvil reflectance and standard deviation derived from one year of satellite observations. There are 18 bins along each dimension, with 5° bin increments from 2.5° to 87.5° (±2.5°) for both VZA and SZA, and 10° bin increments from 5° to 175° (±5°) for RAA, where 0° RAA is the backscattering angle. Figure 3 illustrates the average anvil reflectance for overhead Sun, for a solar zenith angle near 45°, and approaching early sunset. Although the allowable VZA limit for actual observations is ~88.4°,

it should be noted that the maximum observed VZA for this model is ~77.6°, and sampling per VZA bin can suffer with increasing viewing angle in the poleward direction, where deep convection is less likely to be founFu, D.rthermore, because VIS-based observations can become highly shadowed and variable at large SZAs, we caution the use of this method where SZA is greater than 82° despite an allowable LUT limit of up to 90°. Even below this limit, increased sun angle can cause significant shadowing from OT, thereby excluding potential samples (see next paragraph). Therefore, it is not surprising that sample size is

sparser, and uncertainty is higher, at the most extreme angular bins, especially when compounded by both high SZA and VZA. This pattern can be observed in Figs. 4 and 5, which illustrate the bin sampling and reflectance standard deviation ($\sigma$), respectively.

Anvil reflectance pixels must satisfy three homogeneity criteria before being included in the LUT. First, each pixel of a 5 × 5 window/array centered on the pixel of interest must initially be classified as an anvil using the IR-based method (see Appendix

A). As such, each pixel of the 5 × 5 window must have a rating greater than 0, and the rating of the pixel being considered for the LUT (i.e., the center pixel of the 5 × 5 window) must be greater than 75. These requirements help ensure that each included pixel is in an area of reasonably contiguous anvil detection as determined by the IR method, and is therefore likely to be separated from anvil cloud edges where the quality of anvil detection is more suspect due to semitransparency effects. This 5 × 5 array technique must inherently exclude pixels within two spaces of the image edge, but this is necessary given that the continuity of

the anvil cloud beyond the image boundary is unknown, and therefore we are unable to reliably judge those edge pixels.

The IR anvil rating threshold of 75 was chosen for the LUT based on empirical judgement of the relationship between the detection rating and remarkably obvious, spatially coherent cold clouds, which leads to higher certainty in the model. Compared to having no anvil threshold criterion, setting the anvil rating cutoff to 75 reduces the overall dataset size by 6.5% while slightly lowering LUT bin standard deviations, but the magnitude of the LUT does not meaningfully change. Nominally and in this study,



both the IR and VIS anvil masks are defined by an anvil detection rating of 15 or more, which is again a determination made based on empirical assessment of satellite imagery from varied regions and seasons. Having different thresholds for defining a mask and developing the LUT of expected reflectance is acceptable because it is critically important that the LUT observations are exceptionally consistent and predictable. As such, with this first criterion we establish a solid foundation for the BRDF model.

The second and third homogeneity criteria require that the standard deviation of the 5 × 5 VIS reflectance array is less than 3% of the 5 × 5 reflectance average, and that the standard deviation of the 5 × 5 IR BT is less than 1 K. These requirements build upon the first criterion by independently quantifying a standard for homogeneity of the surrounding pixels. The thresholds of 3% and 1 K are well-suited for filtering pixels near anvil edges as a secondary check to the anvil continuity test above. More importantly, perhaps, is the ability of these standard deviation checks to exclude anomalous anvil pixels, particularly those that

may be associated with OT or gravity waves. These features and other irregularities in the anvil generate localized temperature variability and shadowing effects that would not satisfy the filter thresholds. In the VIS case, comparing the standard deviation to the 5 × 5 average rather than the center pixel reflectance helps mitigate the rare occurrence that an abnormally bright center pixel surrounded by dark anvil pixels will satisfy the homogeneity filter, given that the standard deviation in such a case is likely to be less than 3% of the bright pixel but not less than 3% of the array average.

**2.3 Kernel-driven BRDF**

Despite the fact that the three-dimensional LUT approach described in the previous section is relatively simple to construct and computationally fast for estimating anvil reflectance across the spectrum of viewing and illumination angles, it suffers from certain drawbacks. The anisotropy of an Earth target is expected to vary continuously with viewing and solar geometry. The finite discretization of angular bins in the LUT, however, can generate sharp discontinuities in the anvil reflectance between

neighboring angular bins. The non-uniformity in the sample sizes between bins also impairs the smoother transition of reflectance across the bins resulting in the discontinuous patterns that are seen in the reflectance contour lines of Fig. 3, especially at higher SZA. In addition, the LUT approach is unable to define an anvil reflectance for bins without convection.

Here we describe the construction of a kernel-based BRDF model for characterization of anvil top-of-atmosphere reflectance at continuously varying SZA, VZA, and RAA. This approach not only mitigates the discretization discontinuities that are an effect

of the LUT approach, but also fills in the missing intermediate bins. The BRDF is described by a linear superposition of a set of weighting functions, e.g., geometric and optical, that characterize its shape, which is the defining concept of a kernel-driven model (Roujean et al. 1992; Wanner et al. 1995). Despite needing to be flexible enough for application to a variety of inhomogeneous scene types, kernel-driven BRDF models are able to adequately provide description of the anisotropic reflectance of natural surfaces (Hu et al. 1997; Wanner et al. 1997; Hu et al. 1999; Breon and Maignan 2017). The BRDF model

is based on the work of Roujean et al. (1992) that describes a bidirectional reflectance $R$ as a linear sum of three kernels in the following form:

$$R(\theta, \psi, \varphi) = K_0 + K_1 f_1(\theta, \psi, \varphi) + K_2 f_2(\theta, \psi, \varphi), \tag{1}$$

where $\theta$, $\psi$, and $\varphi$ are the SZA, VZA, and RAA, respectively. Expressions $f_1$ and $f_2$ are the model kernels defined as analytical functions of $\theta$, $\psi$, and $\varphi$ that represent the geometric and volume scattering components, respectively. These functions are given

in the forms:

$$f_1(\theta, \psi, \varphi) = \frac{1}{2\pi} [(\pi - \varphi) \cos \varphi + \sin \varphi] \tan \theta \tan \psi - \frac{1}{\pi} (\tan \theta + \tan \psi + \chi) \tag{2}$$

and

$$f_2(\theta, \psi, \varphi) = \frac{4}{3\pi(\cos \theta + \cos \psi)} \left[ \left( \frac{\pi}{2} - \xi \right) \cos \xi + \sin \xi \right] - \frac{1}{3}, \tag{3}$$



where

$$\chi = \sqrt{\tan\theta^2 + \tan\psi^2 - 2\tan\theta\tan\psi\cos\varphi} \qquad (4)$$

and

$$\xi = \cos^{-1}(\cos\theta\cos\psi + \sin\theta\sin\psi\cos\varphi). \qquad (5)$$

Terms $K_0$, $K_1$, and $K_2$ are scene-specific kernel coefficients that are determined using the least-squares solution of the linear BRDF function for a given set of observations. That is, $K_0$, $K_1$, and $K_2$ are derived from the solution to Eq. (1) for the available empirical data and at the certain angular configuration, thereby providing the best fit for the analytical functions and the observed reflectance of each bin. Furthermore, the coefficients are linearly interpolated across adjacent three-dimensional bins in order to yield an even smoother transition of predicted $R$ across continuous angular variation. Finally, the analytical expressions of $f_1$ and $f_2$ are defined such that these terms vanish at nadir viewing and overhead sun conditions. Thus, $K_0$ is the isotropic component representing the overhead sun reflectance at nadir view. An illustration of this model is seen in Fig. 6, using input data shown in Fig. 3. Figures 7 shows the BRDF difference relative to the LUT, and Fig. 8 shows the model uncertainty.

It is noteworthy that the Roujean et al. BRDF model was originally derived for characterizing surface reflectance. However, previous studies have shown that it is also applicable for anisotropic correction of the TOA reflectance over pseudo-invariant ground sites, provided that the atmospheric effect above the sites is repeatable and predictable over time (Angal et al. 2010; Bhatt et al. 2017b). For anvil measurements from GEO imagers, the fact that atmospheric absorption is minimal above anvils and that the GEO imagers have consistent imaging schedules with repeating angular combinations supports the argument for using the kernel-based approach for modeling TOA anvil reflectance (Hu et al. 2004). In this study, $K_0$, $K_1$, and $K_2$ are computed for each angular bin utilizing the satellite-observed reflectance acquired within the bin supplemented by additional measurements from the neighboring bins that are ±5° apart in SZA and VZA, and ±25° apart in RAA from the center bin. Using the extended set of input data for computing the kernel coefficients ensures that the transition of modeled reflectance is uniform across the bins. In addition, this method also allows for modeling of the bidirectional reflectance for the non-filled bins based on the measurements from surrounding bins. The 1-$\sigma$ uncertainty of the modeled reflectance for a given angular bin is defined by the standard error of the regression that is computed for the least-squares fit between the analytical functions and the observed reflectance values during the determination of $K_0$, $K_1$, and $K_2$.

**2.4 Visible anvil mask overview**

As was noted in the Sect. 1, many satellite-based methods have been developed to identify anvil clouds. These methods typically rely on IR and/or WV absorption-band BT and are perhaps augmented by ancillary information such as tropopause temperature from weather prediction models or reanalysis. Bedka and Khlopenkov (2016) demonstrated a method that incorporates spatial analysis for identification of anvil cloud pixels, but this approach is designed to capture anvil regions near OT and not the entire anvil cloud. A COD threshold-based method for recognizing DCC or anvil features, like that described by Hong et al. (2007), can help to address assumptions of full anvil extents, but because the method is pixel-based and does not incorporate spatial analysis it can be adversely impacted by shadowing due to texture or OT. As such, there existed a need for a reliable VIS-based anvil mask as an important prerequisite to the Bedka and Khlopenkov (2016) VIS texture and OT detection algorithm, in that a search for texture and OT should only occur within the accurate full extents of an anvil cloud.

Anvil reflectance prediction using the method described in Sects. 2.2 and 2.3 coupled with spatial analysis offers an opportunity to address previous limitations and enable efficient texture and OT detection. The VIS texture detection process is based on Fourier analysis of spatial frequencies in the VIS imagery. Spatial frequencies consistent with texture are present not only within convection, but also amongst scattered clouds, and especially near cloud edges. The anvil mask is used to limit the Fourier





analysis to only the actual anvil clouds, thereby eliminating false detections associated with other cloud types while also reducing processing time. Classification of the VIS anvil mask is similar to that for the IR mask (Appendix A), except with scoring based on VIS input with reference to the BRDF model.

As described by Bedka and Khlopenkov (2016), the VIS anvil mask is determined by a scoring system that uses an accumulation
of histogram-derived information from a nearby ensemble of pixels relative to the assessed pixel. The input VIS reflectance imagery is processed in subsets described by a 50-km-diameter circular window evaluated at every other pixel and every other line. The peak of the reflectance-based histogram within each subset is evaluated in a way such that the smooth, uniform signature associated with an anvil cloud is detected, which should ideally exhibit a tall and narrow distribution. An initial VIS anvil rating is constructed based on three main considerations: 1) the width and height of the histogram peak, excluding a
possible peak at low-reflectance bins that correspond to clear-sky areas, 2) the difference between the observed reflectance and some nominal reflectance predicted by the BRDF model, and 3) the existence of saturated pixels corresponding to bright OT edges. The second consideration listed above exemplifies the main purpose of this article – marking the major distinction between the VIS anvil mask derivation described here and the original method of Bedka and Khlopenkov (2016), who relied on an empirically-derived function of $\cos(\theta)$ to formulate a nominal anvil reflectance that lacked consideration for bidirectional
effects. The nominal reflectance relative to the histogram maximum prominence determines whether a VIS anvil rating of either 8 or 16 is assigned for the assessed subset pixel and surrounding pixels. As such, anvil rating for a given pixel will accumulate as the window moves through the image, until finally the accumulated mask rating is resampled to the original non-subset resolution (Bedka and Khlopenkov 2016).

The result of the initial considerations is a preliminary mask of pixels corresponding to evenly bright areas, although possibly
with some saturated pixels. The area for inclusion in the mask is then expanded by 6-10 km (larger expansion is used for higher SZA) to include any regions that potentially contain cloud shadows such as those around OT cores, which may have been missed by the histogram analysis. A shadowed pixel is then included in the mask if it is sufficiently surrounded by pixels defined in the preliminary mask. Finally, the expanded mask is multiplied by a scaled difference between the tropopause temperature and the pixel BT. This last step ensures that the resulting anvil mask corresponds to sufficiently cold areas that match the actual extents
of an anvil cloud, and furthermore indicates that there remains an IR component to the VIS mask despite its designation.

### 2.5 Comparison with CloudSat anvil cloud detection

Anvil detections from GOES-16 based on the VIS mask, the IR mask, the WV−IR BTD method, and a tropopause-normalized IR temperature threshold method (i.e., IR−Trop BTD) were validated against independent determinations of anvil clouds from CloudSat using ~800 CloudSat granules from January, April, July, and October of 2018. The CloudSat definition of anvil is
based on the method of Young et al. (2013) and relies on the 2B-CLDCLASS product. Following their technique, an anvil cloud is determined when high/cirriform clouds are connected to within 33 product profiles of a vertical DCC cluster of sufficient depth, provided that the region below the high clouds is cloud free or only partially filled by single-layer, low-level clouds (Young et al. 2013). Receiver operating characteristics (ROC) curves, which report true positive rate against false positive rate, are then determined relating the rate of agreement with CloudSat anvil indications (probability of anvil detection) within 5
minutes using each of the four methods listed above to the rate of false alarms (positive indications that disagree with CloudSat). Note that because of the nature of CloudSat measurements (two-dimensional vertical profiles along the satellite track), these comparisons can only be considered validations in a relative sense, rather than an absolute sense. That is, because the CloudSat profile must encounter a DCC cluster within 33 profiles of high/cirriform clouds in order for the clouds to be assessed as anvils, it is possible that true anvil clouds remain unidentified by CloudSat simply because the DCC cluster associated with them was



not along the scan path. Therefore, false alarm rates may be incorrectly inflated in this validation approach. Also, given the ~13:30 local equator crossing time of CloudSat, these validation results are only representative of low-SZA conditions. Nevertheless, being that all four methods are assessed with these same limitations, we believe their relative comparison remains fair.

## 2.6 Cloud optical depth parameterization

A simple parameterization for anvil COD based on the difference between observed (Obs) VIS reflectance and BRDF-model-predicted anvil reflectance was developed. This feature is important for nowcasting, i.e., flight routing for airborne science campaigns, because it allows for rapid estimation of COD based on readily available input, whereas multi-band cloud retrieval algorithms, such as that of SatCORPS (Minnis et al. 2011; Minnis et al. 2020), require ancillary datasets, additional preliminary

computations, and longer processing time (e.g., cloud masking is required before COD retrieval). That is, this parameterization can approximate imager multi-band-retrieved COD in a matter of seconds rather than minutes, which is significant for real-time weather applications. The parameterization is developed based on the SatCORPS COD product, and thus will introduce additional error on top of the uncertainty of those retrievals. Therefore, the approximation should not be used as a replacement for the true retrievals but rather should be employed as a general estimate of COD when timeliness is the chief concern.

The approximation is defined by an exponential fit of COD as a function of Obs minus BRDF (Obs−BRDF) reflectance, calculated as a function of SZA, with twenty-eight 3° SZA bins from 0°-3° to 81°-84°. Figure 9 shows the 0°-3°, 45°-48°, and 78°-81° fits in order to highlight how the shape of the functional relationship of COD to Obs−BRDF reflectance changes with SZA. The fits are based on calibrated VIS reflectance (Scarino et al. 2017; Doelling et al. 2018) and ice-cloud COD using Minnis et al. (2020) methodology, which was adapted to GOES-16 imagery over the CONUS in July 2018 for pixels coincident with the

VIS anvil mask. Rather than being fit to the entirety of the dataset that satisfies each SZA bin, the two-term exponential model is simply guided by the maximum density of data found along the curve, as indicated by black circles in Fig. 9. Fitting as such prevents influence from outliers and bad retrievals, and therefore better models the most common functional relationship between COD and Obs−BRDF reflectance.

The COD parameterization consistency was evaluated relative to its SatCORPS reference by comparing with GOES-16

COD that is independent from the training dataset, derived from daytime imagery of Hurricane Florence on 11 September 2018. This date was chosen because Florence maintained Category 4 intensity throughout the day and thus sustained a large area of persistent anvil cloud across the full spectrum of SZA, which is unlike land-based convection that typically exist for a few hours in the late afternoon. With land-based convection, it is difficult to distinguish whether variations in COD are due to increasing/decreasing storm intensity, or if they are being caused by a parameterization that is dependent on SZA. It is important

to note that although an intense hurricane should have relatively consistent COD throughout a day when averaged across the storm anvil, variations in reflectance associated with spiral band development or eyewall replacement do occur, which can appear in our data.

## 3  Results and discussion

### 3.1 Features of the kernel-driven model

As was described in Sect. 2.3, the most significant result of the kernel-driven BRDF is the mitigation of discretization discontinuities between adjacent LUT angular bins and completion of bidirectional reflectance for non-filled bins based on measurements from surrounding bins. These effects are apparent, as the patterns seen in Fig. 6 are smoother and more coherent





than those of Fig. 3, with gaps filled, thereby creating a more complete three-dimensional model. Keep in mind that the kernel-driven approach fills gaps in LUT based on an interpolation scheme that draws not only from neighboring VZA and RAA bins, but also from adjacent SZA increments. This is the reason why Fig. 6c, for example, appears exceptionally more continuous than Fig. 3c despite an apparent lack of valid VZA and RAA bins with which to interpolate from to complete the model as shown.

The kernel-driven model allows for filling of angular bins to exactly one bin beyond the valid coverage of the LUT.

Figure 7 reveals, in a qualitative sense, the amount of smoothing accomplished by the kernel model, with the difference pattern highlighting the chaotic nature of the observation-based LUT as owed to sampling inadequacies. The kernel-driven BRDF is largely consistent with the LUT, excepting smoothing differences, at low- and mid-SZA positions (Figs. 7a and 7b, respectively) as indicated by pale purple (0-1% difference) to dark blue (2-3% difference) shading. The largest divergences of the model from

the LUT are at high-SZA positions and where sampling is low. That is, the largest differences shown in Fig. 7b align with areas of low sampling or high uncertainty in Figs. 4b and 5b, respectively. The large differences in Fig. 7c are a result of the greater amount of interpolation required given the discrete nature of the LUT in this volume, which has a much higher associated 1-$\sigma$ uncertainty compared to that of the lower SZA conditions (compare Fig. 5c to Figs. 5a and 5b), and also overall higher uncertainty than that of the model (Fig. 8). Note that the 1-$\sigma$ uncertainty of Fig. 5 is based on the actual samples and average of

each LUT bin, whereas in Fig. 8, recall that 1-$\sigma$ uncertainty comes from the standard error of the regression computed for the least-squares fit between the analytical functions and the observed reflectance values, which benefits from the incorporation of measurements from surrounding bins and thus has lower uncertainty for each bin compared to that of the same bin from the LUT, assuming the LUT bin is filled. Despite the uncertainties in high-SZA conditions, the kernel-driven BRDF model exemplifies significant improvements over the simple LUT in terms of anvil characterization because continuous, smooth transitions across

bin thresholds lends to a more realistic pattern of anvil reflectance.

## 3.2 Anvil mask comparisons

A robust anvil mask based on VIS interpretation should perform similarly regardless of perceived changes in cloud brightness owed to viewing or illumination conditions, which is the motivation behind the BRDF model. This premise was introduced earlier with Fig. 1, where the same cloud structures were viewed simultaneously with either GOES-East or GOES-West in the

forward-scatter position. Figure 10 revisits that imagery, but now with the VIS anvil mask (Sect. 2.4) determined from either GOES-East or GOES-West indicated with a red line. In either case (12:30 UTC or 23:45 UTC), the general shapes of the masks are comparable despite rather large apparent differences in the calibrated reflectance values, especially in the 12:30 UTC example (Figs. 10a and 10b). Fine-scale differences in the shapes of the red line are present but overall the masks are in agreement.

Another way to qualitatively evaluate anvil mask performance is through comparison with other IR-based methods described above. Examples of these methods are provided in Fig. 11, which shows developing thunderstorms near Kansas and Missouri observed by GOES-16. Figure 11a overlays the IR anvil mask (Appendix A), which had been used to construct the VIS anvil reflectance database for the LUT, and the VIS anvil mask described above. The VIS mask identifies the bright DCC whereas the IR mask extends beyond the boundaries of the VIS mask into pixels that are still cold but not as bright. Even in the northeast

portion of the image where the explanation for disagreement is perhaps more questionable, close examination reveals that the VIS mask is outlining a narrow region of bright cloud whereas the IR mask identifies less cohesive shapes. Both approaches define the anvil, and the preferable approach depends on application and tolerances of those that may use the data. Commercial aircraft, for instance, may choose to avoid any indication of anvil cloud out of an abundance of caution. In contrast, airborne





research field campaign mission planners, such as those during the High Ice Water Content – Radar missions (Bedka et al. 2019), were interested in only the coldest and most reflective clouds.

The VIS and IR masks aim for self-consistency and reliability but with broad applicability, independent of changes in time, location, or satellite source. The drawbacks of other anvil identification methods were mentioned in Sect. 1, some of which are

highlighted and expanded upon in Figs. 11b-11d. For instance, Fig. 11b showcases an example of why DCC or anvil characterization based solely on simple IR BT thresholds is problematic. A 205-K threshold, which is suitable for the tropics, performs poorly for this CONUS scene. In this case, a 225-K threshold is generally more consistent with the VIS and IR mask results.

Rather than using a simple threshold, one can normalize IR BT by the MERRA-2 tropopause temperature reanalysis (i.e.,

"TROPT" from the "inst1_2d_asm_Nx" dataset), which accounts for latitudinal dependencies in cloud top height/temperature and is interpolated in time and space to the satellite pixels. Figure 11c shows the difference of GOES-16 IR BT and the MERRA-2 tropopause temperature (IR−Trop BTD). In terms of anvil detection, it is unclear what IR−Trop BTD threshold would perform best. A BTD < 10 K provides results consistent with the VIS and IR masks. Less-positive BTD values will gradually restrict the mask to only smaller areas in and around OT regions, whereas greater positive values can easily yield false detections.

The WV−IR BTD method for anvil detection is also demonstrated (Fig. 11d). Aside from the similar question of which BTD best defines anvil clouds, there is a more practical limitation of this technique to consider. Because the WV spectral response functions vary across imagers, WV−IR BTD uniformity across multiple sensors is difficult to achieve. Furthermore, the signal-to-noise ratio is low for cold WV BT, and the fairly broad WV channels limit the WV−IR BTD detection of anvil clouds to the most extreme cases (Ai et al. 2017). Also, having a WV channel is not guaranteed for common sensors, whether historical

instruments such as AVHRR or for newer instruments such as VIIRS. Therefore, although the WV−IR BTD approach is reliable and relatively accurate, it is not a consistently available option. As such, given that the VIS and IR mask techniques rely only on two standard channels, the methods are more portable and dependably applicable to past, current, and future platforms.

Figure 12 offers a more quantitative comparison of the relative performance of each anvil identification technique with respect to the CloudSat reference (see Sect. 2.5). Probability of anvil detection as defined by CloudSat, or agreement of each identification

method with CloudSat, is related to the false alarm rate, or occurrences when a method identifies an anvil but CloudSat does not. Naturally, as the definition of what qualifies as an anvil becomes more restrictive by each method's standards (e.g., VIS mask anvil defined by 1 or greater vs. 20 or greater, or WV−IR BTD anvil defined by -8 K or greater vs. 0 K or greater), the probability of detection decreases. Similarly, less restrictive definitions expectedly result in greater false alarm rates. Therefore, in an ideal case, the perfect anvil definition would maximize probability of detection while minimizing false alarm rate and

would thereby be situated as near as possible to the top left corner of the ROC graph. In the context of this relative comparison, the VIS mask offers comparable or slightly better probability of anvil detection and lower false alarm rate than any of the other three methods. That is, overall the four methods are quantifiably similar in effectiveness. The VIS mask, however, when applicable in low-SZA conditions, has perhaps a slight advantage in anvil designation. For high SZA or VZA conditions, in which shadowing can be prevalent, IR-centric methods are preferred.

### 3.3 Hurricane Florence cloud optical depth parameterization

Morning (10:52 UTC top row), mid-day (16:42 UTC center row), and evening (21:47 UTC bottom row) GOES-16 observed reflectance (left column), COD derived from Obs−BRDF anvil reflectance (center column), and COD from SatCORPS (right column) values for Hurricane Florence are shown in Fig. 13. These morning, mid-day, and evening views of Florence have average SZAs of about 79.1°, 23.8°, and 79.8°, respectively. The SZA values are computed from the mean SZA within the red





contour, which signifies the 34-kt wind radii (one radius per quadrant for a total of four 34-kt radii). The NOAA National Hurricane Center provides wind quadrant radii for maximum sustained wind values of 34, 50, and 64 kts. The coordinates defining each quadrant extend in the NE, SE, SW, and NW directions, radiating outward from the center of the storm to a distance where the indicated windspeed is expected to be possible. This study uses these radii as a basis for COD evaluation,

with each one drawn on the Fig. 13 imagery in red (34 kts), magenta (50 kts), and blue (64 kts).

In the morning, the high SZA creates shadows on Florence cloud tops due to OT and gravity waves, seen most prominently near the eyewall but also within the outflow shield (Fig. 13a). Compared to mid-day (Fig. 13d), the overall observed reflectance is lower at around 0.7 to 0.8 on average, rather than ≥ 0.9 when the Sun is higher overhead. According to the BRDF model for this morning angular combination (Fig. 14), the predicted anvil reflectance is between 0.71 and 0.81 – gradually increasing from the

southwest to the northeast part of the image. The COD derived by the SZA-dependent functional relationship (Fig. 9), namely Fig. 13a minus Fig. 14, is shown in Fig. 13 b. The areas where observed reflectance is much less than predicted reflectance is where low COD values are expected as signified by the darkest shades. The veracity of these dark shades within the outflow shield is questionable, however, because it is unlikely that COD is fluctuating so rapidly across these short distances where deep convective clouds are located. In other words, shadowing and texture generates severely under-estimated COD within three-

dimensional cloud-top structures. This pattern, however, is consistent with SatCORPS results (Fig. 13c), which is our current baseline for comparison as the parameterization is dependent on the SatCORPS reference. Bedka et al. (2019) show that reflectance smoothing prior to COD computation dampens shadowing effects at high SZA, which results in a more spatially consistent product. That smoothing approach, however, was purposefully excluded for this study. Observed reflectance that is only slightly less than predicted values signifies a greater COD, shown in gray shades. Cloud optical depth grows exponentially

as the observed reflectance matches and surpasses the predicted anvil reflectance. Note that despite the potential for the exponential function to predict excessive COD values given rather modest changes in either reflectance value, COD is capped at a maximum value of 150 to be consistent with SatCORPS output.

At high SZA, the exponential growth of COD with Obs−BRDF is rather extreme and therefore uncertainty is high. Based on Fig. 9c, when Obs−BRDF is close to 0, a combined 0.05 error in observed or predicted reflectance could amount to the difference

between 70 and 150 COD, or an 80-COD unit variance. At such early and late times in the day (as Figs. 13g, 13h, and 13i behave similarly to 13a, 13b, and 13c) the function is steep and therefore highly sensitive to reflectance uncertainty, not to mention the increased standard error of the fit itself due to variable SatCORPS retrievals at these high-SZA conditions. On the other hand, at midday (Figs. 13d, 13e, and 13f), the exponential function is less steep, with a shape somewhere between that of Figs. 9a and 9b. Here the BRDF only varies between 0.91 and 0.94 across the entire image (even less across the wind radii). A combined 0.05

error close to where Obs−BRDF is 0 in this case results in about 10-COD variance, and with lower standard error of the fit compared to the previous case. Exactly how impactful 80-COD error would be in near-sunset applications, or 10-COD error during midday operations, is dependent on the product application. For a simple and immediate means of estimating the broad-scale COD conditions, however, this method performs well relative the more computationally intensive, although likely overall more accurate, COD retrieval method.

The Obs−BRDF parameterized COD is compared to SatCORPS COD within the 34-, 50-, and 64-kt radii of Hurricane Florence using the daytime (SZA < 82°) 5-minute imagery from GOES-16 on 11 September 2018. The COD results as a function of UTC hour for each radii set are shown in Fig. 15, with Obs−BRDF COD in red and SatCORPS COD in blue. The SZA is also displayed above the x-axis of each plot. Overall, the Obs−BRDF COD agrees rather well with SatCORPS throughout the day, which is reassuring given that the parameterization was developed using independent SatCORPS COD data. The mean COD

differences between Obs−BRDF and SatCORPS (Obs–BRDF minus SatCORPS) for the 34-, 50-, and 64-kt radii are 3.7%, 0.7%,





and, 1.1%, respectively. The agreement with SatCORPS is encouraging, especially near 80°-82° SZA where COD differences are around 11 at worst and 0 at best, because it validates a consistency in approach that is independent of viewing and illumination conditions, at least to the extent that SatCORPS is similarly independent, which is the purpose of the BRDF model.

## 4. Summary

Operational forecasting of severe and aviation weather, especially in regions without adequate contiguous weather radar coverage, can benefit significantly from rapid and highly detailed imaging offered by geostationary satellite measurements. Consistent imagery-based identification of severe weather indicators, such as deep convective updrafts, anvil clouds, and OT, can be difficult to achieve, however. This article highlights a kernel-driven BRDF model for informed prediction of anvil reflectance, which helps anvil cloud detection efforts and ultimately improves the two-channel, passive satellite imager OT detection algorithm. A satellite VIS-based detection algorithm that incorporates predicted anvil reflectance for known angular conditions is able to more consistently identify DCC anvils on the scale afforded by satellite imagery, regardless of viewing and solar conditions, which is beneficial to a variety of stakeholders.

The kernel-driven BRDF model, which is described by a linear superposition of a set of geometric and optical weighting functions, is employed to characterize the anvil top-of-atmosphere reflectance at continuously varying SZA, VZA, and RAA. This approach effectively mitigates discretization discontinuities and fills missing intermediate bins in the LUT, thereby creating a reliable model for predicted anvil reflectance. Despite lingering uncertainties at high-SZA positions, the kernel-driven BRDF model improvement over the LUT is significant because continuous, smooth transitions across discrete angular bins results in a more natural pattern of predicted reflectance, which benefits anvil characterization efforts.

The VIS mask is a more conservative approach in anvil detection than the initial IR mask. The masks may disagree on what strictly defines the limit of an anvil, but either offers advantages depending on the application. These techniques are also independent of geographic considerations, unlike methods based on static IR thresholds, and are not as susceptible to false positives that arise due to ill-defined BTD allowances such as is the case for IR−Trop-based anvil detection. The WV−IR BTD technique offers reliable, well-documented performance in anvil classification, but consistent identification across the entire constellation of geostationary satellites is not guaranteed. Therefore, a two-channel approach based on widely available VIS and IR imagery grants broader applicability and dependability for well-calibrated imagers. Furthermore, in a relative comparison study with a CloudSat anvil reference, the VIS anvil mask offered better skill in anvil identification for low-SZA conditions than any of the IR-centric methods.

By subtracting BRDF-model-predicted anvil reflectance from observed VIS reflectance we are able to develop a simple parameterization for anvil COD. An SZA-dependent exponential fit of SatCORPS-derived COD as a function of Obs−BRDF reflectance defines the parameterization, which produces an approximation of SatCORPS COD but with significantly less computational demand. The exponential growth of COD with Obs−BRDF is rather extreme at high SZA, and thus COD is sensitive to small changes in observed or predicted reflectance. Regardless, for a simple and immediate means of estimating the broad-scale COD conditions that is at least comparable to SatCORPS to within 2% on average, this parameterization works well.

### Acknowledgments

This work was supported by the NASA Applied Sciences Disasters Program under award number 18-DISASTER18-0008. We thank Douglas Spangenberg (SSAI at NASA LaRC) for assisting with the data processing. The angular-dependent kernel coefficients are available upon request to those looking to compute specific bidirectional reflectance values.



## Appendix A: Calculating the IR anvil mask

The steps below describe how spatial infrared temperature patterns are quantified to identify convective anvil clouds in the form of an "anvil mask." The anvil mask is a rating that indicates a confidence in anvil detection, with values above 10-15 roughly corresponding to human perception of anvil cloud extents and values above 100 indicating a high level of confidence.

Quantification of anvil confidence is a crucial step in development of the kernel-driven BRDF.

Pixel-level IR BT data are first subtracted from the local tropopause temperature in order to obtain the brightness temperature difference (BTD) relative to the tropopause. This BTD is processed in circular subsets of 22-km diameter extracted at every other pixel and every other line. The local distribution of BTD within each subset is analyzed by constructing a histogram $H$ having $N=32$ bins and covering the range from -35 K (i.e., warmer than the tropopause), which is a low tropopause-relative bound for

anvil clouds, to 13 K colder than the tropopause, which only occurs in updraft regions. Pixels colder than the 13-K threshold are accumulated in the last histogram bin. Figure A1 shows examples of BTD histograms calculated for a typical anvil cloud within a convective system (red columns) and for a region outside the anvil cloud (blue columns).

The BTD within anvils should exhibit spatially uniform cold temperature values, which in most cases will result in a sharply peaked histogram. As such, it follows that anvil rating should be made proportional to the peak height $H_i$, which indicates the

number of counts in the $i$-th bin, and therefore is also proportional to that bin's index $i$ given that higher bin indices correspond to colder regions, e.g., Fig. A1. The following formula, refined through extensive testing, describes the dependence of anvil rating $r_{anvil}$ on index $i$:

$$r_{anvil} = \frac{0.35}{D^2} \cdot H_i \cdot i \cdot (2N + 8 - i). \tag{A1}$$

Here, $D$ is the diameter of the histogram window in pixels and the term in parentheses acts to gradually flatten the $r_{anvil}(i)$

dependence at higher levels of confidence in anvil detection as BTD reaches zero and becomes strongly positive. Based on the example shown in red in Fig. A1, using an 11-pixel diameter $D$ at 2-km pixel resolution, the peak in the red histogram at bin 20 has a height of 23, which yields an IR anvil rating of 69. In most cases the formula above describes uniformly cold anvil clouds reasonably well. If non-uniform regions around OT cores are causing the histogram peak to split over several bins, the major peak can be counted together with neighboring bins to make the total contribution equivalent to a single strong peak, thereby

lending stability in resultant $r_{anvil}$ across the whole anvil.

Finally, the obtained anvil mask has to be expanded in order to include pixels along the anvil boundary, where there is only partial anvil coverage in the subsetting window. This expansion is implemented by raising the rating for all anvil pixels inside the 22-km circular window that have BTD larger than 7.5 K below the histogram's peak. Their anvil rating is increased to reach the level of the peak bin. After this spatial expansion, the IR anvil mask presents a reasonable match relative to the actual anvil

cloud region, with the anvil extents filled with the nearly uniform field of $r_{anvil}$.

Although this method attempts to identify uniform cloud areas near the tropopause, it does not guarantee that a broad area of extremely cold cloud is indeed an anvil cloud. For example, a large area of cold jet stream cirrus in a winter storm may be assigned a significant anvil rating if a local histogram happens to have sufficient criteria. This leaves some room for improving the IR anvil rating, for instance by 1) incorporating a difference relative to the regional background in order to help define

convective environments (i.e., cold cloud vs. much warmer clear sky background), or 2) using model-derived atmospheric instability indices, such as convective available potential energy, to restrict detections to regions where deep convection is assumed to be possible. Nevertheless, practical experience with developing the mask, the graphical examples shown in this paper, and comparisons with CloudSat indicate that the IR anvil mask performs reasonably well and is suitable for constructing the BRDF model.



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

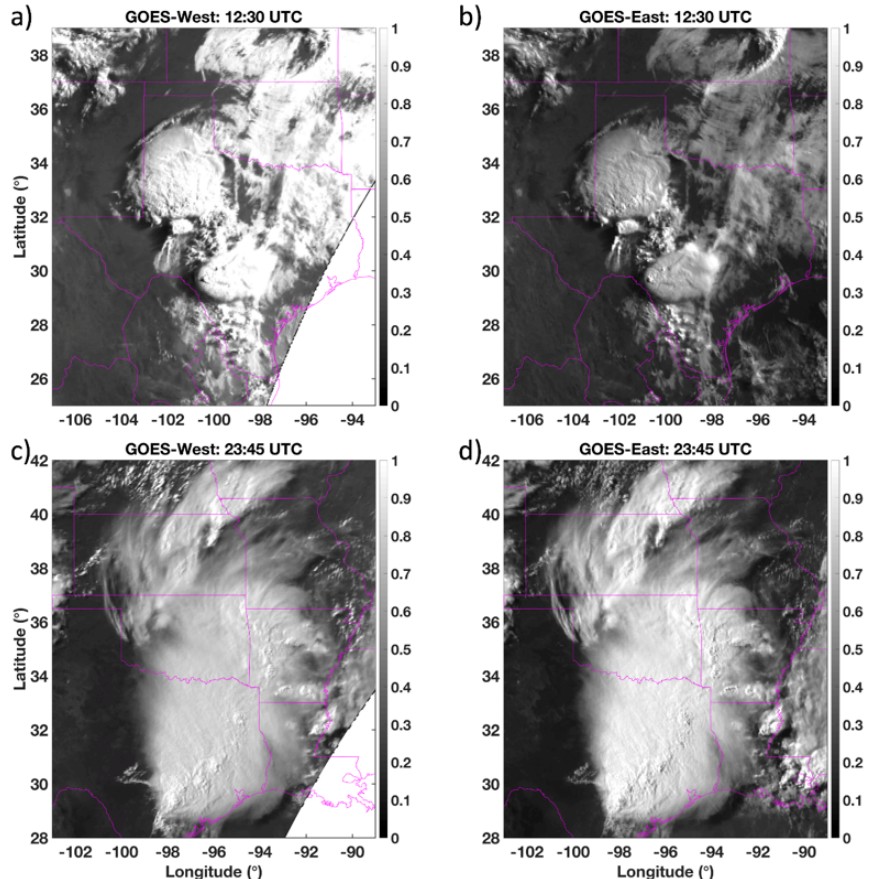

**Figure 1: Calibrated VIS reflectance of a 25 May 2015 MCS over Texas and Oklahoma as viewed by a) GOES-West (GOES-15) at 12:30 UTC, b) GOES-East (GOES-13) at 12:30 UTC, c) GOES-West at 23:45 UTC, and d) GOES-East at 23:45 UTC. The apparent brightness of the MCS changes as the solar illumination and viewing conditions vary.**

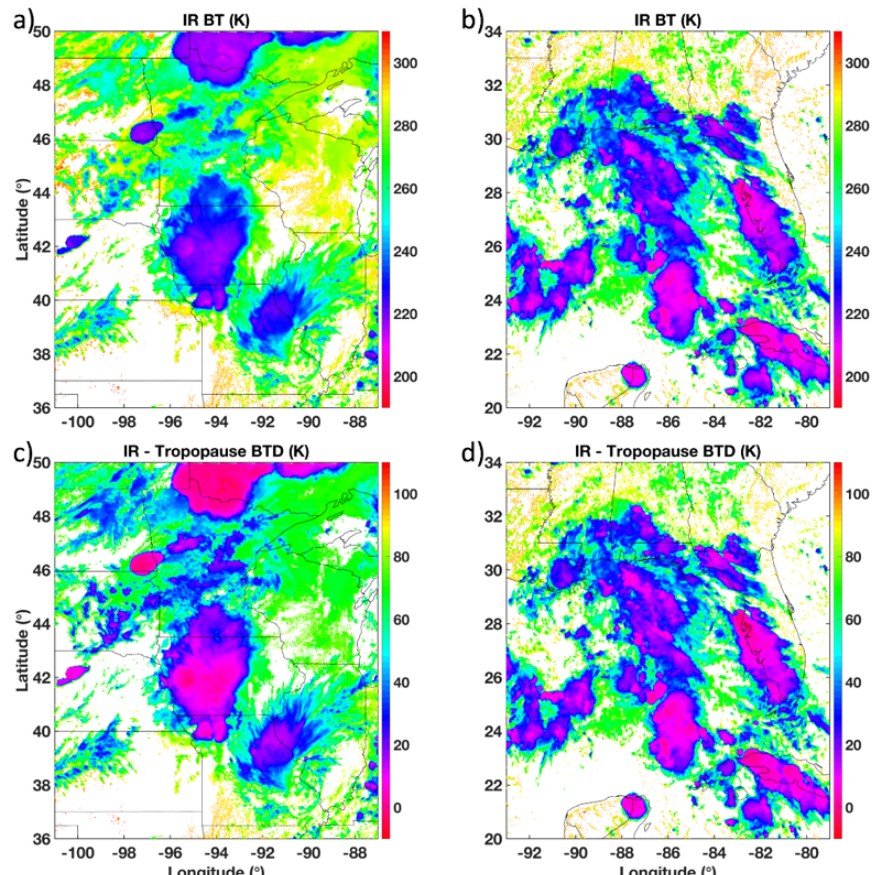

**Figure 2: Severe storms on 31 August 2018 at 20:47 UTC visualized by GOES-16 a) IR BT at northern CONUS latitudes, b) IR BT at southern CONUS latitudes, c) IR−Trop BTD at northern CONUS latitudes, and d) IR−Trop BTD at southern CONUS latitudes. Without tropopause normalization, the relative intensity of the northern storms appears less than that of the southern storms despite both being significant producers of severe weather.**

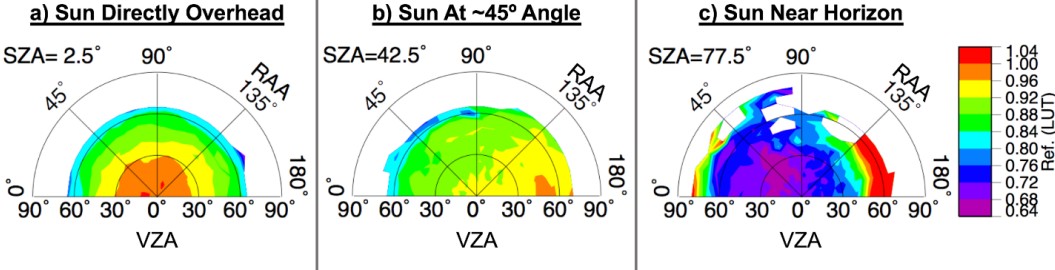

**Figure 3: Illustration of LUT for average anvil reflectance as a function of VZA, SZA, and RAA, based on 2016 December through 2017 November GOES-13, GOES-15, and Himawari-8 retrievals. Polar plots are shown for the a) 2.5º, b) 42.5º, and c) 77.5º SZA bins (±2.5°). The radial coordinates of each plot indicate the change in VZA, demarcated into 18 bins with 5° bin increments from 2.5° to 87.5° (±2.5°). The polar coordinates of each plot indicate the change in RAA, demarcated into 18 bins with 10° bin increments from 5° to 175° (±5°), where 0° RAA is the backscattering angle. Gaps at certain bin indices indicate a lack of anvil sampling for that angular configuration.**





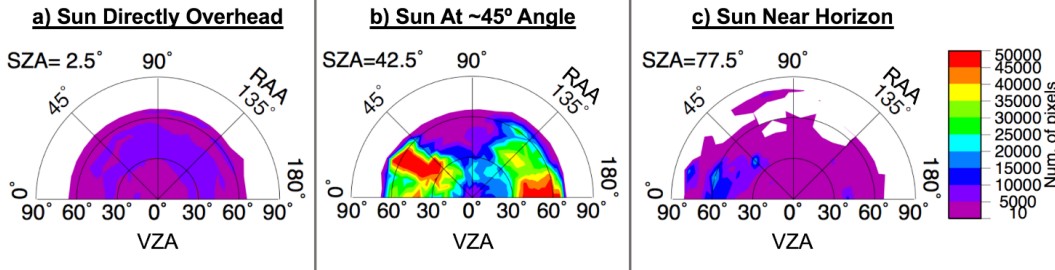

Figure 4: Sampling distribution for the LUT shown in Fig. 3.





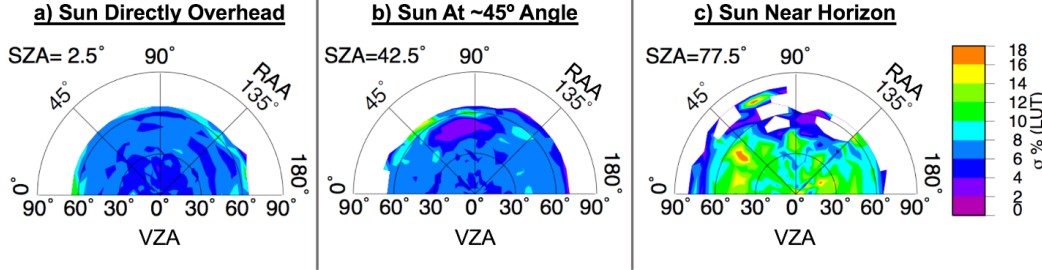

**Figure 5: Uncertainty of the LUT shown in Fig. 3 given as the standard deviation percentage of the average bin reflectance.**





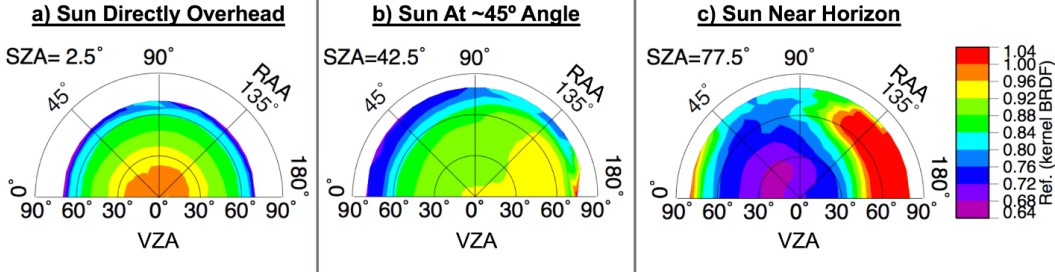

**Figure 6: A semi-empirical kernel-based BRDF model of anvil top-of-atmosphere reflectance at continuously varying SZA, VZA, and RAA. The coordinate system is the same as that described in Fig. 3.**





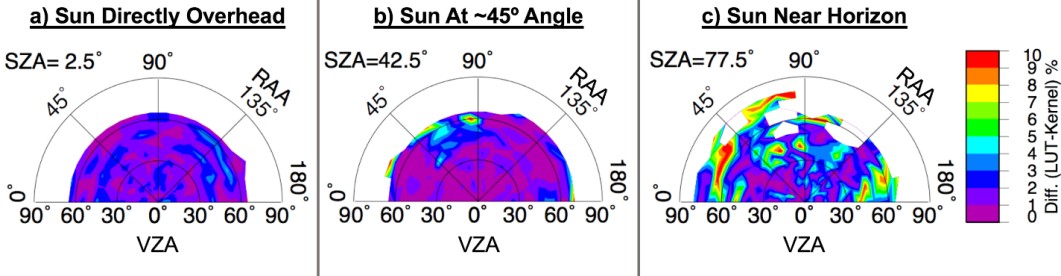

**Figure 7: Percent difference in predicted anvil reflectance between the LUT shown in Fig. 3 and the kernel-based BRDF model shown in Fig. 6.**





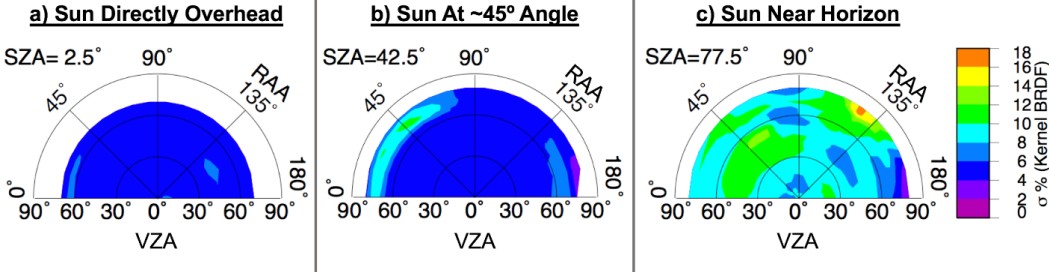

**Figure 8: Uncertainty of the kernel-based BRDF model shown in Fig. 6 based on the standard error of the regression computed for the least-squares fit between modeled and observed reflectance values.**





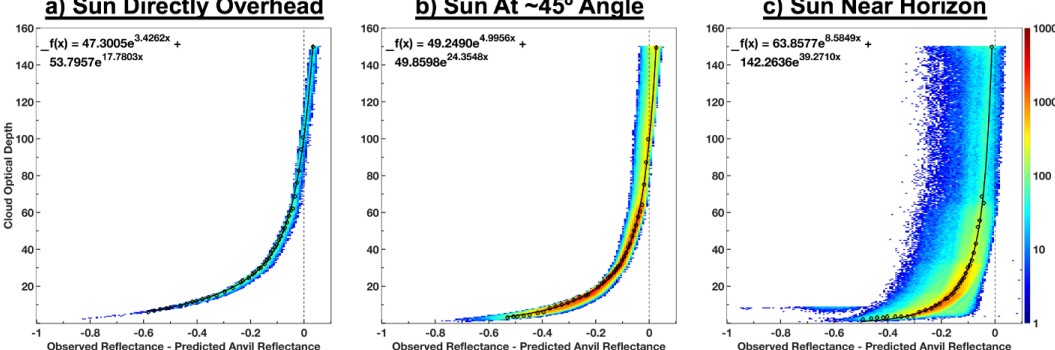

**Figure 9: Cloud optical depth as a function of SatCORPS Obs−BRDF based on July 2018 CONUS retrievals for SZA ranges of a) 0º to 3º, b) 45º to 48º, and c) 78º to 81º.**

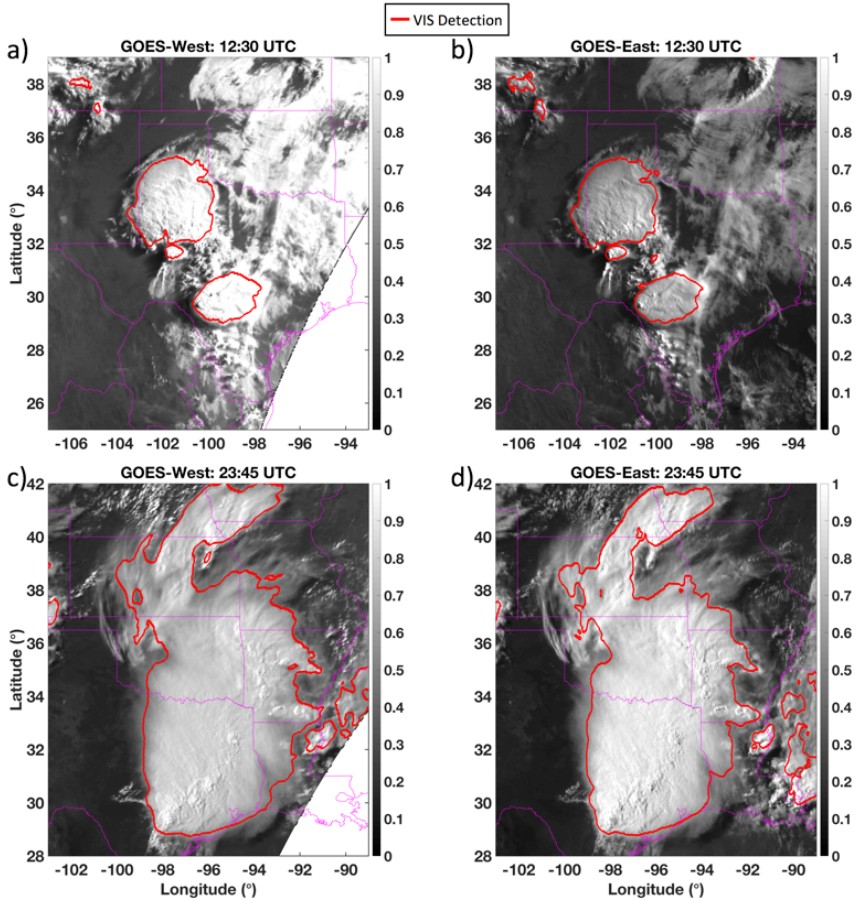

**Figure 10: Same as Fig. 1 except with the GOES-West or GOES-East VIS anvil mask outlined in red. The masks from each satellite are similar at the corresponding times despite extreme viewing and illumination differences.**



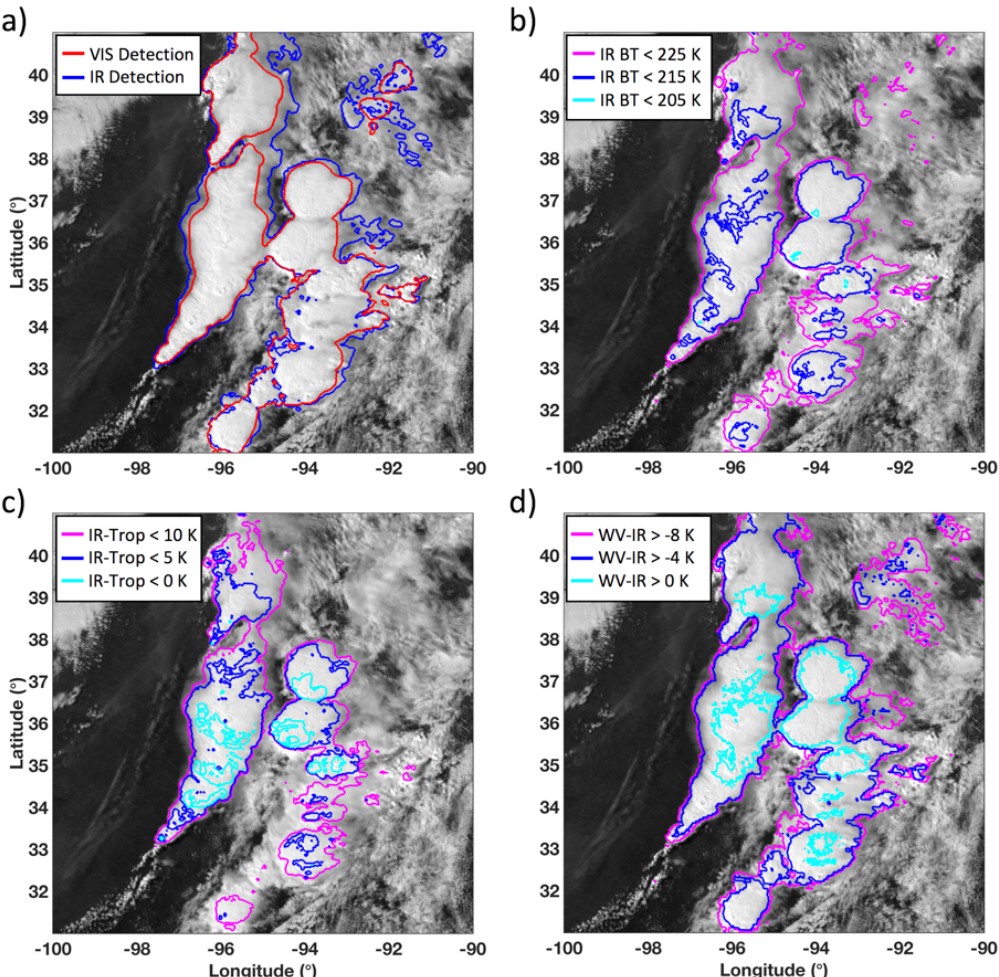

**Figure 11: A cluster of developing thunderstorms near Kansas and Missouri on 13 April 2018 viewed from GOES-16 at 20:45 UTC overlaid with a) the VIS and IR anvil masks, b) contours for IR BT less than 225, 215, and 205 K, c) contours for IR−Trop BTD less than 10, 5 and 0 K, and d) contours for WV−IR BTD greater than -8, -4, and 0 K.**



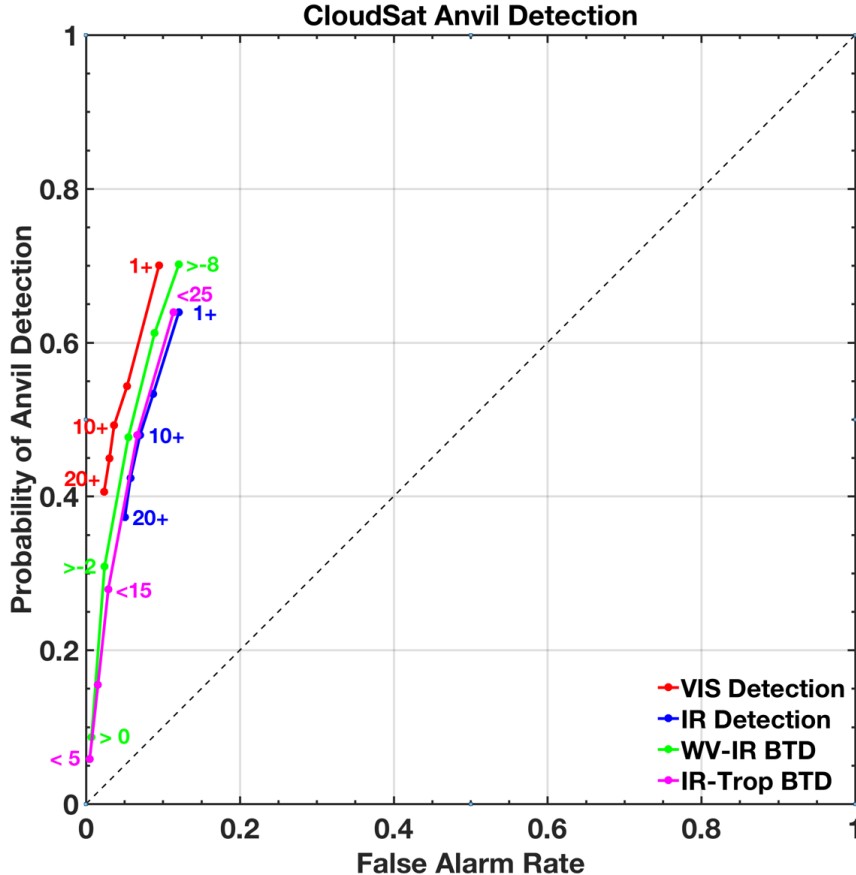

**Figure 12: Receiver operating characteristic curves highlighting the rate of CloudSat anvil detection based on the VIS mask, the IR mask, the WV−IR BTD test, and the IR−Trop BTD test relative to the rate of false alarms. The VIS and IR masks are evaluated from ratings of 1 or greater up to ratings of 20 or greater, the WV−IR BTD test is evaluated from differences of -8 K or greater to 0 K or greater, and the IR−Trop BTD test is evaluated from differences of 25 K or less to 5 K or less.**



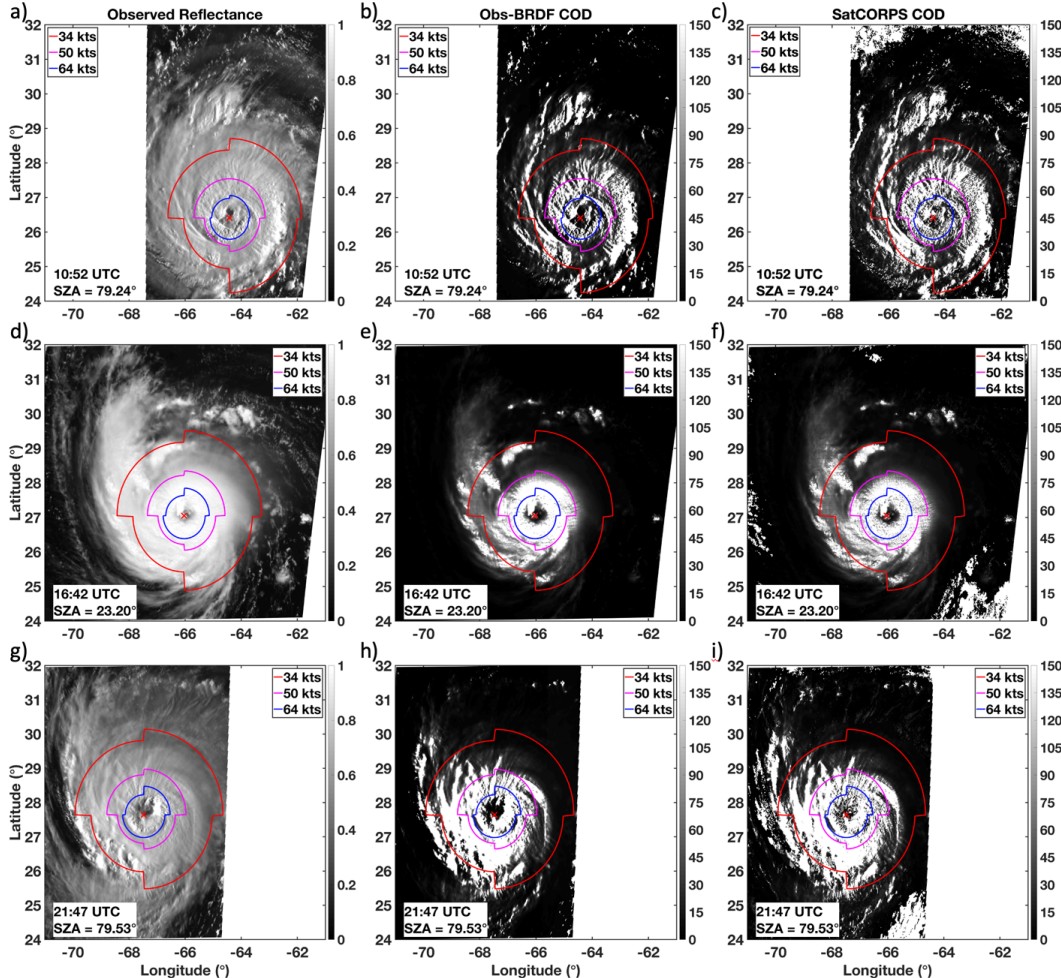

**Figure 13: Observed VIS reflectance (left column), Obs−BRDF COD (center column), and SatCORPS COD (right column) imagery of Hurricane Florence from GOES-16 on 11 September 2018 at 10:52 (top row), 16:42 (center row), and 21:47 UTC (bottom row). Wind radii contours provided by the NOAA National Hurricane Center are indicated at 34, 50, and 64 kts, and the storm center is marked with a red 'X.' The average SZA within the 34-kts radii is displayed at the bottom of each panel. White areas beyond the edge of the image are either unprocessed parts of the domain or are regions with SZA greater than 82º.**

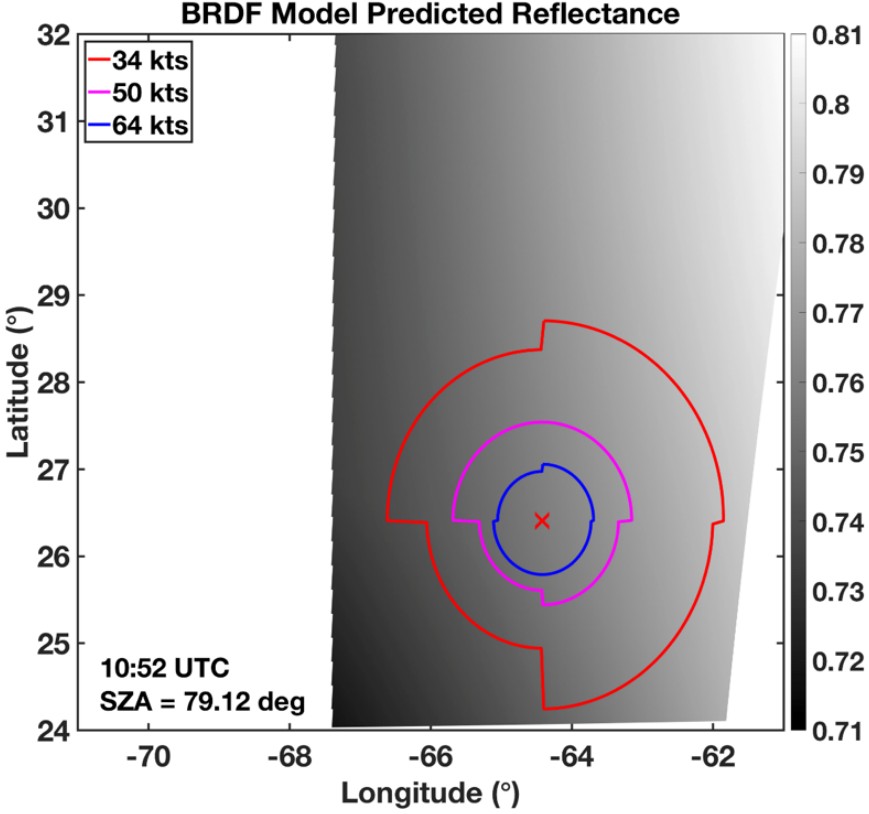

Figure 14: Predicted anvil reflectance based on the kernel-driven BRDF and GOES-16 viewing/illumination geometry at 10:52 UTC on 11 September 2018 over Hurricane Florence. White areas beyond on the edge of the image are either unprocessed parts of the domain or are regions with SZA greater than 82°.

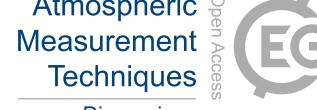

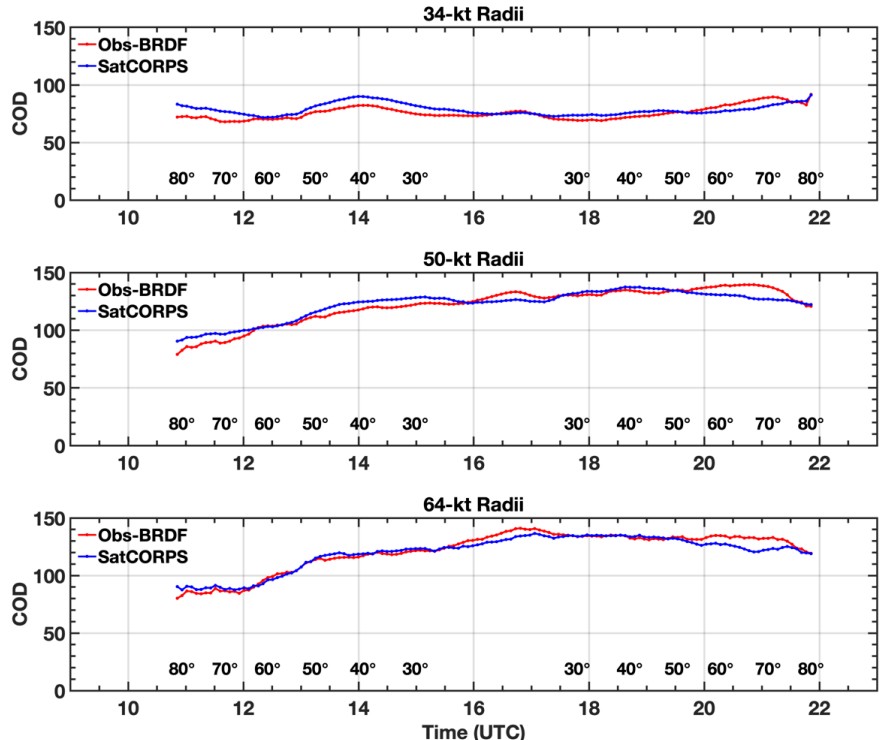

Figure 15: Cloud optical depth as a function of time (UTC hour) and SZA (listed in degrees) from SatCORPS (blue) and as determined from the SZA-dependent Obs−BRDF function (red) as shown in Fig. 9, derived over Hurricane Florence on 11 September 2018 within the 34-, 50-, and 64-kt radii as shown in Figs. 13 and 14.





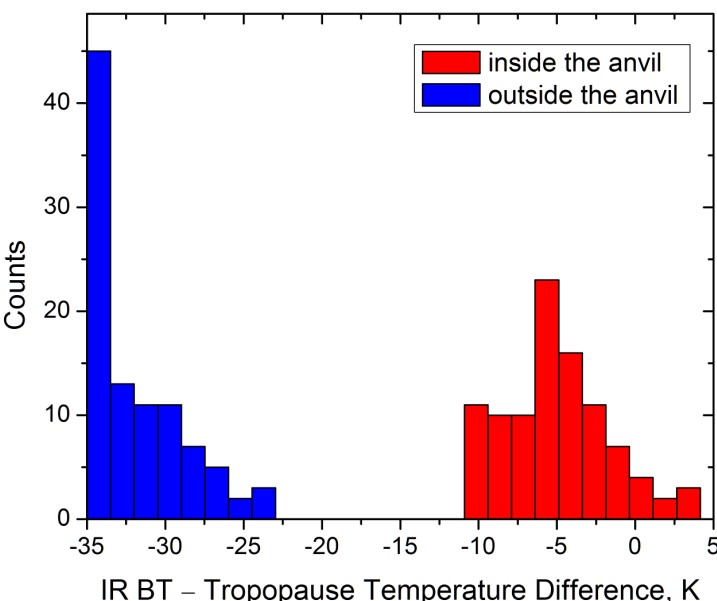

**Figure A1: BTD histograms calculated over two regions observed by GOES-16 on 5 May 2019 at 23:00 UTC: inside a typical anvil cloud (red) and outside the anvil cloud (blue). On the red histogram, the peak is identified at bin 20 equaling 23 counts.**