# Peer review of "A kernel-driven BRDF model to inform satellite-derived visible anvil cloud detection"

_Atmospheric Measurement Techniques, 2020_

## Referee Comment (RC1) · Anonymous Referee #1 · 29 Jun 2020

**General comments**

In general, quantitative uses of the visible bands on the GOES are under-utilized. This work will greatly help with that issue, to better automatically detector and characterize anvil clouds.

**Specific comments**

Page 1. Line 28. Consider these missing references: Brunner, J. C., S. A. Ackerman, A. S. Bachmeier, and R. M. Rabin, 2007: A quantitative analysis of the enhanced-V feature in relation to severe weather. Wea. Forecasting, 22, 853–872. McCann, D. W., 1983: The enhanced-V: A satellite observable severe storm signature. Mon. Wea. Rev., 111, 887–894.

[Figure]

Please reference Ted Fujita's "jumping (stratospheric) cirrus": Fujita, T. T., 1974: Over-shooting thunderheads observed from ATS and Learjet. Satellite and Mesometeorology Research Project Rep. 117, Texas Tech University, Lubbock, TX, 29 pp.

Menzel, W. P., and J. F. W.Purdom, 1994: Introducing GOES-I: The first of a new-generation of Geostationary Operational Environmental Satellites. Bull. Amer. Meteor. Soc., 75, 757–781.

Page 2. Line 5. It should be noted that the legacy GOES (eg, GOES-13/15) were calibrated pre-launch, but not on orbit. Unless, you have applied a X. Wu (NOAA STAR) visible calibration correction (to account for degradation over time). If this is the case, it should be stated. Details of the correction: https://www.star.nesdis.noaa.gov/smcd/spb/fwu/homepage/GOES_Imager_Vis_OpCal.php Is this what the authors are referring to, when they refer to "spectral band normalization"?

Page 2. Line 29. Note also that the brightness temperature of a water vapor band varies with view angle. For example, a cooling of over 5K for the legacy GOES imager water vapor band at 55 Local (Viewing) Zenith Angle: http://cimss.ssec.wisc.edu/goes/calibration/GOES12_IMGR_LZAvsTEMP.jpg

Page 4. Line 26. "The VIS retrievals are subsampled to the IR data resolution". Was an averaging approach considered? Or ran on one day to understand if there are any differences between sampling or averaging?

Page 5. Line 3. It's stated that Meteosat aren't used due to a lack of 1km vis data over full disk, but the 1km is sampled to 4km in your analysis anyway. So, would using 3km vs 4km be that different? Especially considering that the 3km is at the subpoint? Isn't a larger difference with Meteosat is the timing, given those data are scanned south to north, while the GOES scan north to south?

Page 5. Line 10. Please clarify how this relates to the paragraph: "Note that for some

analyses, satellite data are supplemented by modeled atmospheric profiles provided by the Global Modeling and Assimilation Office (GMAO) Modern-Era Retrospective analysis for Research and Applications, Version 2 (MERRA-2) product."

Page 6. Line 5. How was the 5x5 determined? Given that a projected field-of-view size grows away from nadir, this mean that less area is considered near the sub-point compared to near the limb. How might this affect your results?

Page 8. Line 6. This is every other 4km line and 4km element, correct? If so, this should be stated here, for clarity.

Page 8. Line 11. Do you screen out the sun glint times/location?

Page 9. Line 19. When it is stated "adapted to GOES-16 imagery", does this still mean using 4km spatial resolutions for both the vis and IR inputs? If so, consider running a case (for comparison) at 2km, to use more of both the vis and IR information. Add a GOES-16 reference: http://nwafiles.nwas.org/jom/articles/2018/2018-JOM4/2018-JOM4.pdf

Page 9. Line 25. Consider adding a GOES-16 COD reference.

Page 11. Line 17. Even more importantly, water vapor BT varies strongly with few angle, much more than an IR longwave window.

Page 13. Line 6. This might be a good spot to reference: Line, W. E., T. J. Schmit, D. T. Lindsey, and S. J. Goodman, 2016: Use of Geostationary Super Rapid Scan Satellite Imagery by the Storm Prediction Center. Wea. Forecasting, 31, 483–494, https://doi.org/10.1175/WAF-D-15-0135.1.

Page 14. Line 8. Again, is the every other line/element based on 4km pixels? If so, this should be stated for clarity.

Figure 1. Define what calibration means. Pre-launch? Adjusted for degradation over time? State that the images have been re-mapped to a common projection.

Figure 2. Severe storms based on NOAA storm reports of hail? Wind?

Technical corrections

Page 5. Line 22. Typo: founFu, D.rthermore

###

---

## Referee Comment (RC2) · Martin Setvak (Referee) · 3 Jul 2020

**Referee Comments – Martin Setvák, CHMI**

03 July 2020

**General Comments**

The paper focuses on automatic detection of anvils of deep convective clouds (DCC), based on BDRF model developed by the authors. Various anvil detection techniques or products are being used in (satellite-based) nowcasting systems, thus any new similar method can enhance credit of the satellite data. This gains on importance with recent onset of new generations of GEO satellites, such as Himawari-8/9, GOES-R series, FY-4A series, or GEO-COMPSAT-2A, or the upcoming third generation of Meteosat satellites (MTG). For these reasons I welcome the submitted paper, and recommend it for publication.

**Specific Comments**

Page 2, Lines 25 – 29

I would be somewhat more conservative about usefulness of the WV-IR BTD method, namely for the overshooting tops (OT) detection. It depends not only on availability of appropriate WV channel and scanning geometry, but for specific cases namely on presence and total amount of water vapor in the lower stratosphere, above the storms, and its vertical thermal profile. Reading this part as it is written now may impose an impression that this technique is broadly used for OT detections, being reliable – which is far from the reality. However, I do not dispute its use for detection of DCC in general.

Page 4, Lines 1 – 2

As written, *its application in enhancing anvil cloud detection (and thereby OT detection) capability*, it may seem that the method can be directly used for OT detection. Though the authors elaborate this statement later in the paper, perhaps a more accurate wording might help here.

Page 4, Lines 16 – 17

Can there be any impact of the location of the satellites – Himawari-8 providing data namely for DCC above the ocean, while GOES satellites depicting namely storms above the continent?  I'm not speaking here about different underlying surface, but rather about different types and concentrations of the condensation nuclei above continent and oceans, which may affect the cloud top microphysics and thus also its reflectance (BRDF) …

Page 14, Line 13 and 24

… *should exhibit spatially uniform cold temperature values* … You discuss here the impact of colder overshooting tops, but how about the enclosed warm areas of storms exhibiting cold-Vs or cold rings? How does the algorithm deal with these?

**Other comments**

I can hardly discuss the technical details of this work (as I have no personal experience in this area), however from an observational perspective and long-term personal experience with satellite observations of storm tops, the individual steps, their settings and parametrization seem to be reasonable and justified. I hope that the authors plan extension (or verification) of this work also to the GOES-16 and GOES-17 data, and possibly also to Meteosat's SEVIRI and future FCI data.

---

## Author Comment (AC1) · 25 Aug 2020

General comments

In general, quantitative uses of the visible bands on the GOES are under-utilized. This work will greatly help with that issue, to better automatically detector and characterize anvil clouds.

**Thank you for taking the time to review and offer feedback for our paper. Our responses are offered below, and any applicable changes are indicated by page and line number.**

Specific Comments

Page 1. Line 28. Consider these missing references: Brunner, J. C., S. A. Ackerman, A. S. Bachmeier, and R. M. Rabin,2007: A quantitative analysis of the enhanced-V feature in relation to severe weather. Wea. Forecasting, 22, 853–872. McCann, D. W., 1983: The enhanced-V: A satellite observable severe storm signature. Mon. Wea. Rev., 111, 887–894.

Please reference Ted Fujita's "jumping (stratospheric) cirrus": Fujita, T. T., 1974: Overshooting thunderheads observed from ATS and Learjet. Satellite and Mesometeorology Research Project Rep. 117, Texas Tech University, Lubbock, TX, 29 pp.

Menzel, W. P., and J. F. W.Purdom, 1994: Introducing GOES-I: The first of a newgeneration of Geostationary Operational Environmental Satellites. Bull. Amer. Meteor. Soc., 75, 757–781.

**Thank you for these suggestions, the references have been included.**

Page 2. Line 5. It should be noted that the legacy GOES (eg, GOES-13/15) were calibrated pre-launch, but not on orbit. Unless, you have applied a X. Wu (NOAA STAR) visible calibration correction (to account for degradation over time). If this is the case, it should be stated. Details of the correction:
https://www.star.nesdis.noaa.gov/smcd/spb/fwu/homepage/GOES_Imager_Vis_OpCal.php Is this what the authors are referring to, when they refer to "spectral band normalization"?

**We apologize for not making this clear initially, but the "calibrated visible reflectance" we're discussing here and elsewhere in the paper is neither referring to the pre-launch nor to Fred Wu's on-orbit calibration corrections. Rather we are referring to CERES Edition 4 visible imager calibration method coefficients of Doelling et al. 2018, which are determined from the monthly gain trends of GEO and Aqua MODIS spectrally consistent, ray-matched radiance**

pairs over all-sky tropical ocean, deep convective clouds, and deserts. The spectral consistency, or spectral band normalization, aspect of the calibration is ensured by the application of spectral band adjustment factors (SBAFs) derived by convolving scene-specific hyperspectral data from the SCIAMACHY instrument with the spectral response functions of the reference (MODIS) and target (GEO) sensors, further described by Scarino et al. 2017. Although we did reference these two articles and pointed to the Aqua MODIS standard shortly after the initial mention of *calibrated visible reflectance* and *spectral band normalization* (originally Page 2, Line 9), further explanation about the calibration origin did not come until Page 4, starting at Line 34. Also, the SBAF was not specifically mentioned.

**To address this, we have added mention of the CERES Edition 4 calibration method and spectral band adjustment factors to Page 2, starting at Line 5 where the term *calibrated* is first introduced.**

Page 2. Line 29. Note also that the brightness temperature of a water vapor band varies with view angle. For example, a cooling of over 5K for the legacy GOES imager water vapor band at 55 Local (Viewing) Zenith Angle:
http://cimss.ssec.wisc.edu/goes/calibration/GOES12_IMGR_LZAvsTEMP.jpg

**Thank you for this note. We have added discussion and reference of the strong VZA dependency in the WV band for legacy GOES starting at Page 2, Line 30 and Page 11, Line 38.**

Page 4. Line 26. "The VIS retrievals are subsampled to the IR data resolution". Was an averaging approach considered? Or ran on one day to understand if there are any differences between sampling or averaging?

**Your question reveals that we did not clearly explain how we handle resampling, subsampling, their distinction, and what data resolutions are utilized at the different steps of the process. For a short explanation, the key points are: 1) There is generally little resultant difference from subsampling vs. averaging in regard the method being described, which is aggregation of anvil reflectance, given the homogeneity of the scene. 2) Resampling to a fixed grid, subsampling, and averaging are all occurring, but the exact process depends on the imager and the variable being output. 3) Retrieval processes do in fact take advantage of the higher VIS resolution. That is, VIS downscaling does not happen until the final output. The full explanation is as follows, which is summarized in the restructured section starting at Page 4, Line 30:**

**We have explored DCC reflectance measurement sensitivity to subsampling vs. averaging and have found that the difference is insignificant for scenes that are relatively homogeneous across many kilometers, which is the case for the anvil clouds we are targeting in this study. The largest deviations between the two methods come in the vicinity of cloud edges, within a field of broken clouds, near OT, or amongst gravity waves. Cloud edges and scattered clouds are easily avoided with IR- and VIS-based homogeneity filters, threshold checks, and continuity assurances regardless of sampling method. Even smaller-scale VIS variabilities**

caused by subtle OT and gravity wave features are filtered comparably whether employing subsampling or averaging because they had first been damped through the 6 × 6 array Lanczos-filtering-based resampling process. The point is that the final reflectance aggregation is comparable whether subsampling or averaging because large homogeneities are discovered either way, and subtle homogeneities have been muted through the resampling process.

That said, we still must explain what method was employed. The text originally did not make it clear that the Lanczos resampling process was also performed on GOES data – not just on Himawari-8. We also did not adequately convey that the Himawari-8 data were first subsampled (by skipping every other line and element using McIDAS-X software) to GOES "1-km" resolution before resampling. The final result of the resampling is a fixed-scale grid at 1-km resolution for VIS, and 4-km resolution for IR. As such, hereafter, unless otherwise stated, the term *pixel* refers specifically to individual data samples of the Lanczos-interpolated fixed grid rather than the actual imager pixel data. Given the two fixed-grid scales, algorithm processes involving VIS can take advantage of 1-km data. Only upon output are the data sampled to 4 km. The method of sampling in this step, which is with respect to the fix-grid, depends on the variable being output. In the case of reflectance, the sixteen 1-km pixels of the corresponding 4-km IR pixel are averaged. In the case of one of our product outputs like visible texture rating, the maximum value is output (although with Fourier smoothing applied – such details that are beyond the scope of this paper but rather are being prepared for an ATBD). The important point is that as far as VIS output is concerned, our method of creating the anvil BRDF of expected reflectance relies only on the resampled reflectance, which prior to output was downscaled to IR resolution by averaging.

It is important to point out that these examples are specific to the satellite data being discussed in this section for the particular purpose of aggregating homogeneous reflectance measurements coincident with the IR anvil mask across legacy GEOs. For a different application we may utilize full-resolution Himawari-8 or GOES-16, in which case VIS would be resampled to a fixed 0.5-km grid, IR to a fixed 2-km grid, algorithm processes would consider 0.5-km VIS data, and final output would be 2-km for all parameters. In this case, sixteen 0.5-km reflectance pixels would be averaged for each 2-km IR box, or, again in the case of visible texture, the maximum value of sixteen would be output.

Page 5. Line 3. It's stated that Meteosat aren't used due to a lack of 1km vis data over full disk, but the 1km is sampled to 4km in your analysis anyway. So, would using 3km vs 4km be that different? Especially considering that the 3km is at the subpoint? Isn't a larger difference with Meteosat is the timing, given those data are scanned south to north, while the GOES scan north to south?

It is correct that that main reason we did not use Meteosat data for the construction of the anvil BRDF (or more precisely, the look lookup table of average anvil reflectance) is because rather than having 1-km VIS imagery across the full field of view as with GOES and Himawari, 1-km MSG VIS imagery is broadly available only for the Northern Hemisphere. In the

Southern Hemisphere, 1-km MSG VIS follows the Sun, which is disadvantageous for an angular dependency model. As described above, we resample VIS reflectance to the Lanczos-interpolated 1-km fixed grid, and then average to a 4-km fixed grid for the purposes of harmonized output with IR. Therefore, the GOES and Himawari processes utilize 1-km VIS, whereas comparable MSG processes would have to use 3-km. The question then is whether we expect a difference in anvil identification and reflectance measurements between Lanczos-interpolated 1-km GOES/Himawari VIS data and 3-km MSG VIS data. For IR we would not expect a significant brightness temperature variation for 3 km vs 4 km given the homogeneous clouds we are targeting. For VIS, however, small-scale OT shadowing and texture variances are likely to yield differences in identification and measurement for 1 km vs 3 km. Also, regarding the point about the 3-km MSG resolution being at the sub-satellite point, the resampling process to achieve the fixed grid masks the effect of FOV growth as view moves from nadir to limb. More discussion on this, however, can be found in response to a comment further below.

The opposite full disk scanning pattern compared to GOES and Himawari would not be a significant concern for the purpose discussed in this section, which is construction of the multi-angle LUT for anvil reflectance. There is no time matching involved in this process, as the only dependency being investigated here is that on combined VZA, SZA, and RAA. That is, we collected half-hourly reflectance measurements independently for each satellite and averaged all results into a single angular-dependent LUT without requirement for time consideration. Convective processes of each satellite region are distinct enough (spatially and over a 1-year period) that inconsistency in scan direction should not introduce systematic bias in reflectance measurements for specific angular configurations.

Page 5. Line 10. Please clarify how this relates to the paragraph: "Note that for some analyses, satellite data are supplemented by modeled atmospheric profiles provided by the Global Modeling and Assimilation Office (GMAO) Modern-Era Retrospective analysis for Research and Applications, Version 2 (MERRA-2) product."

The intent was to conclude Section 2.1 with mention of the MERRA-2 inclusion. We agree, however, that it did not fit naturally as the final sentence of the MSG discussion paragraph. Therefore, we have moved the sentence to the end of the first paragraph of Section 2.1 (Page 4, Line 27), where we feel its description of being a satellite data supplement fits more intuitively.

Page 6. Line 5. How was the 5x5 determined? Given that a projected field-of-view size grows away from nadir, this mean that less area is considered near the sub-point compared to near the limb. How might this affect your results?

The 5x5 array was chosen for homogeneity assessment as the compromise between the model having higher uncertainty or being excessively constrained. First of all, we chose an odd number like 5 rather than 4 or 6 because doing so simplifies evaluation of a centered pixel relative to its surrounding pixels, which are equidistant on the fixed grid. We initially

constructed the LUT based on a 3x3-array homogeneity evaluation, but this allowed for inclusion of questionable anvil pixels given that there was less we could quantitatively interpret in order to filter the results. Our LUT uncertainty (Fig. 5) was significantly higher than what is shown for the 5x5 array case, especially at the higher SZA bins. Aggregating many questionable pixels would not be helpful for a model designed to provide expected anvil reflectance. On the other hand, it is expected that choosing a 7x7 array for evaluation of homogeneity would be too conservative. Spatial coherency and consistency in positive identification from an empirical perspective are the key aspects we aim to capture in our anvil and OT detections products, and thus we believe homogeneity evaluation on a 7x7-pixel scale would exclude too many clouds that an informed person would otherwise identify as anvil.

We don't specifically account for the FOV growth, but it is effectively masked by the resampling process. Of course, the effect does not simply disappear because we resample. Distortion of an input pixel will in some form persist through to the resampled pixel, and the share of satellite pixels being combined in the resampling process is dependent on the viewing angle. Nevertheless, for the purpose of anvil reflectance aggregation, resampling an enlarged satellite pixel from an extreme viewing angle is not detrimental because the FOV is less likely to feature subtle shadowing and fine-scale texture. It is true then that OT detection is negatively impacted by FOV stretching, but that is beyond the scope of this paper. The main importance is that we construct a model that provides the expected reflectance for a given view, regardless of whether extreme views are distorted or are masking features that would otherwise result in exclusion at less extreme angles. In other words, what the imager "sees" at a particular angular configuration is the truth we are aiming to model.

Page 8. Line 6. This is every other 4km line and 4km element, correct? If so, this should be stated here, for clarity.

This is actually in reference to the Lanczos-interpolated fixed VIS grid. The actual numbers are dependent on the satellite input. For example, for a stand-alone GOES-16 VIS anvil mask computation, the resampled 0.5-km VIS reflectance imagery is subset to 1-km by taking every other row and column of the fixed grid. We hope that the edits made in Section 2.1 (explained above) provide better context for this statement. We have also added specific mention of the subsampling being in reference to the fixed grid, and have provided example scale values so that there is no ambiguity (Page 8, Line 21).

Page 8. Line 11. Do you screen out the sun glint times/location?

No, we do not employ any special handling of sun glint regions because any anvil reflectance enhancement owed to glint would be predicted by the BRDF model. Thus, observed reflectance should closely match the nominal reflectance, and anvil determination can continue. It is possible the pixels will be saturated – an occurrence that is specifically considered when the anvil mask is being constructed because OT edges can appear very bright in certain scattering configurations. This is the type of saturation being discussed on

**the line you indicated. Unlike in the BRDF construction, the goal when defining the anvil mask is to be inclusive of anomalous shadowed or saturated pixels that are otherwise surrounded by anvil. So rather than screening out sun glint, the aim is to correctly classify glint-saturated pixels. We have updated the text to include mention of not only bright OT edges, but also sun glint (Page 8, Line 29).**

Page 9. Line 19. When it is stated "adapted to GOES-16 imagery", does this still mean using 4km spatial resolutions for both the vis and IR inputs? If so, consider running a case (for comparison) at 2km, to use more of both the vis and IR information. Add a GOES-16 reference: http://nwafiles.nwas.org/jom/articles/2018/2018- JOM4/2018-JOM4.pdf

**We were attempting to explain that the COD retrieval methodology used for CERES instruments, which is described by Minnis et al. (2020), has been adapted to the SatCORPS Framework as applied to GOES-16, which is nominally run at 4-km resolution over CONUS. Note that the sensitivity of SatCORPS ice cloud COD to retrieval resolution was investigated by Minnis et al. 2016 ([https://www1.ncdc.noaa.gov/pub/data/sds/cdr/CDRs/AVHRR_Cloud_Properties_NASA/AlgorithmDescription_01B-30b.pdf](https://www1.ncdc.noaa.gov/pub/data/sds/cdr/CDRs/AVHRR_Cloud_Properties_NASA/AlgorithmDescription_01B-30b.pdf)), who found no variation in COD whether retrieving at 1-km, 2-km, or 4-km resolution. So to be clear, "adapted to GOES-16 imagery" specifically concerns adaptation of the CERES COD method to SatCORPS GOES-16. That said, the VIS anvil mask that we aligned with the SatCORPS GOES-16 COD for this section on COD parameterization was from a run done with 0.5-km VIS, which hopefully we have now provided better context for given our previous responses and edits. We have changed this sentence to clarify that CERES COD was adapted to SatCORPS GOES-16, referenced the COD retrieval sensitivity analysis, and provided specific scale values for the datasets involved in this process. Thank you for the GOES-16 reference, we have included that as well.**

Page 9. Line 25. Consider adding a GOES-16 COD reference.

**We have performed a comparison with the NOAA GOES-R Series Advanced Baseline Imager Level 2 Cloud Optical Depth (ABI-L2-COD). Using only "good_quality_qf" pixels from ABI-L2-COD, geo-spatially aligned with SatCORPS COD for the same analysis period and regions, the NOAA GOES-16 COD reference is on average ~8% greater than SatCORPS and the parameterized COD, even when seemingly normalized to the same scale. This is a large difference and we feel that further investigation is merited, especially in regard to the nuances of scale normalization and handling of saturated pixels, but such efforts are more suitable for a separate paper that is meant to emphasize product comparisons and validations. The goal of the parameterization, after all, is to quickly estimate SatCORPS COD, which has been accomplished. Exploration of the absolute agreement of SatCORPS COD relative to NOAA ABI-L2-COD is beyond the scope of this work. We did, however, add a GOES-16 COD reference (Page 9, Line 27).**

[Figure]

Page 11. Line 17. Even more importantly, water vapor BT varies strongly with few angle, much more than an IR longwave window.

**We have added discussion and reference of the strong VZA dependency in the WV band for legacy GOES, with specific mention of its significance in the context of historical consistency and comparison to IR window bands (Page 11, Line 38).**

Page 13. Line 6. This might be a good spot to reference: Line, W. E., T. J. Schmit, D. T. Lindsey, and S. J. Goodman, 2016: Use of Geostationary Super Rapid Scan Satellite Imagery by the Storm Prediction Center. Wea. Forecasting, 31, 483–494, https://doi.org/10.1175/WAF-D-15-0135.1.

**Thank you for the reference, we have included it (Page 13, Line 30).**

Page 14. Line 8. Again, is the every other line/element based on 4km pixels? If so, this should be stated for clarity.

**As before, this in reference to the Lanczos-interpolated fixed grid, but this time for the IR resolution. Also as before, the actual scale values are dependent on the application and satellite input. E.g., for a GOES-16 IR anvil mask computation, the resampled 2-km IR BT imagery is subset to 4-km by taking every other row and column of the fixed grid. For the application of anvil reflectance aggregation across GOES-13, GOES-15, and Himawari-8, as this paper describes, the IR anvil mask was developed from the 4-km fixed-grid IR BT, subsampled to 8-km resolution. We have changed the sentence to now offer both of these examples (Page 14, Line 32).**

Figure 1. Define what calibration means. Pre-launch? Adjusted for degradation over time? State that the images have been re-mapped to a common projection.

**We now define the reflectance as inter-calibrated with reference to MODIS, with further explanation provided in the text. We also now state that the images were remapped to a common projection.**

Figure 2. Severe storms based on NOAA storm reports of hail? Wind?

**We changed the caption to start as, "Severe wind- and hail-producing storms (as reported by NOAA)…"**

Technical corrections

Page 5. Line 22. Typo: founFu, D.rthermore

**Typo corrected.**

---

## Author Comment (AC2) · 25 Aug 2020

**Referee Comments – Martin Setvák, CHMI**

03 July 2020

General Comments

The paper focuses on automatic detection of anvils of deep convective clouds (DCC), based on BDRF model developed by the authors. Various anvil detection techniques or products are being used in (satellite-based) nowcasting systems, thus any new similar method can enhance credit of the satellite data. This gains on importance with recent onset of new generations of GEO satellites, such as Himawari-8/9, GOES-R series, FY-4A series, or GEO-COMPSAT-2A, or the upcoming third generation of Meteosat satellites (MTG). For these reasons I welcome the submitted paper and recommend it for publication.

**Thank you for your time and insight in reviewing our manuscript. You will find our responses to your comments below with indicated changes in the text where applicable.**

Specific Comments

Page 2, Lines 25 – 29
I would be somewhat more conservative about usefulness of the WV-IR BTD method, namely for the overshooting tops (OT) detection. It depends not only on availability of appropriate WV channel and scanning geometry, but for specific cases namely on presence and total amount of water vapor in the lower stratosphere, above the storms, and its vertical thermal profile. Reading this part as it is written now may impose an impression that this technique is broadly used for OT detections, being reliable – which is far from the reality. However, I do not dispute its use for detection of DCC in general.

**We agree with your assessment. Although Ai et al. (2017) did demonstrate capability for WV-IR BTD to detect OT, it was neither reliable nor the main focus of their BTD to noise ratio, which rather was DCC detection. Also, given that our manuscript does not directly concern OT, it is probably better to not mention OT in this discussion. Therefore, we have removed the sentence that started, "Ai et al. (2017)… " originally appearing on Page 2, Line 26. We have also muted our emphasis on WV-IR BTD reliability in general (Page 12, Line 4 and Page 14, Line 8) and introduced the fact that WV bands on legacy GOES have very strong VZA dependency (Page 2, Line 30 and Page 11, Line 38)**

Page 4, Lines 1 – 2
As written, *its application in enhancing anvil cloud detection (and thereby OT detection) capability*, it may seem that the method can be directly used for OT detection. Though the authors elaborate this statement later in the paper, perhaps a more accurate wording might help here.

**In order to better convey that the BRDF model does not directly enhance OT detection, we have adjusted the sentence to read, "… its application in enhancing anvil cloud detection capability and cloud optical depth (COD) parameterization" (Page 4, Line 3).**

Page 4, Lines 16 – 17
Can there be any impact of the location of the satellites – Himawari-8 providing data namely for DCC above the ocean, while GOES satellites depicting namely storms above the continent? I'm not speaking here about different underlying surface, but rather about different types and concentrations of the condensation nuclei above continent and oceans, which may affect the cloud top microphysics and thus also its reflectance (BRDF) …

**True there are regional differences in DCC reflectance owed to different microphysics, but they should not have a significant impact on our results. Doelling et al. (2018) measured a 0.8% difference in the count response between the TWP region and the Meteosat region from ~2003-2007, which is the largest regional difference they observed. This corresponds to ~0.9% difference in reflectance and is not enough to have meaningful impact on predicted nominal reflectance from the BRDF model.**

Page 14, Line 13 and 24
… *should exhibit spatially uniform cold temperature values* … You discuss here the impact of colder overshooting tops, but how about the enclosed warm areas of storms exhibiting cold-Vs or cold rings? How does the algorithm deal with these?

**The warm areas of such features are evaluated in the same way as the rest of the anvil using the 22-km moving window. These portions are not warm enough to negate detection, but they are likely to be assigned lesser IR anvil ratings than the colder portions. As far the anvil BRDF is concerned, enclosed warm areas would likely be excluded from the reflectance LUT aggregations because they would not pass the BT homogeneity test. These rare exclusions, however, should not significantly influence the nominal reflectance predicted by the BRDF model, and therefore the final VIS anvil mask results would not be affected. We have added these details to the text on (Page 15, Line 17).**

Other comments
I can hardly discuss the technical details of this work (as I have no personal experience in this area), however from an observational perspective and long-term personal experience with satellite observations of storm tops, the individual steps, their settings and parametrization seem to be reasonable and justified. I hope that the authors plan extension (or verification) of

this work also to the GOES-16 and GOES-17 data, and possibly also to Meteosat's SEVIRI and future FCI data.

**Thank you again for your comments. Extension of the BRDF model of expected anvil reflectance is planned to follow the launch of MTG.**